# A Ce-CuZn catalyst with abundant Cu/Zn-O$_V$-Ce active sites for CO$_2$ hydrogenation to methanol

Runping Ye [1,11], Lixuan Ma [2,11], Jianing Mao [3,4,11], Xinyao Wang [5], Xiaoling Hong [5], Alessandro Gallo [6], Yanfu Ma [5], Wenhao Luo [7], Baojun Wang [2], Riguang Zhang [2] ✉, Melis Seher Duyar [8,9] ✉, Zheng Jiang [10] ✉ & Jian Liu [5,7,8] ✉

CO$_2$ hydrogenation to chemicals and fuels is a significant approach for achieving carbon neutrality. It is essential to rationally design the chemical structure and catalytic active sites towards the development of efficient catalysts. Here we show a Ce-CuZn catalyst with enriched Cu/Zn-O$_V$-Ce active sites fabricated through the atomic-level substitution of Cu and Zn into Ce-MOF precursor. The Ce-CuZn catalyst exhibits a high methanol selectivity of 71.1% and a space-time yield of methanol up to 400.3 g·kg$_{cat}^{-1}$·h$^{-1}$ with excellent stability for 170 h at 260 °C, comparable to that of the state-of-the-art CuZnAl catalysts. Controlled experiments and DFT calculations confirm that the incorporation of Cu and Zn into CeO$_2$ with abundant oxygen vacancies can facilitate H$_2$ dissociation energetically and thus improve CO$_2$ hydrogenation over the Ce-CuZn catalyst via formate intermediates. This work offers an atomic-level design strategy for constructing efficient multi-metal catalysts for methanol synthesis through precise control of active sites.

Carbon dioxide (CO$_2$) hydrogenation can be a major enabling technology for establishing a carbon neutral circular economy. Chemicals and fuels such as methanol, light olefins, and gasoline can be produced through CO$_2$ hydrogenation[1–4]. Methanol can be used as liquid fuel for transportation, serving as hydrogen carrier for renewable energy to chemicals schemes[5]. Additionally, methanol is a useful solvent and feedstock for synthesis of other chemicals such as olefins[6,7]. Thus, the

implementation of CO$_2$ hydrogenation to methanol could not only reduce CO$_2$ emission but also generate useful products.

Cu-based catalysts have been widely used for chemical synthesis due to their low-cost and high-performance under intermediate reaction temperatures[8–10]. The commercial methanol synthesis process is based on converting a synthesis gas feed (a mixture of CO, CO$_2$ and H$_2$ typically obtained from fossil fuels) over a Cu/ZnO/Al$_2$O$_3$ catalyst[6].

[1]Key Laboratory of Jiangxi Province for Environment and Energy Catalysis, Institute of Applied Chemistry, School of Chemistry and Chemical Engineering, Nanchang University, Nanchang 330031, PR China. [2]State Key Laboratory of Clean and Efficient Coal Utilization, College of Chemical Engineering and Technology, Taiyuan University of Technology, Taiyuan 030024 Shanxi, PR China. [3]Shanghai Institute of Applied Physics, Chinese Academy of Sciences, Shanghai 201204, PR China. [4]Center of Materials Science and Optoelectronics Engineering, University of Chinese Academy of Sciences, Beijing 100049, PR China. [5]State Key Laboratory of Catalysis, Dalian Institute of Chemical Physics, Chinese Academy of Sciences, Dalian 116023 Liaoning, PR China. [6]SUNCAT Center for Interface Science and Catalysis, SLAC National Accelerator Laboratory, 2575 Sand Hill Road, Menlo Park, CA 94025, USA. [7]College of Chemistry and Chemical Engineering, Inner Mongolia University, Hohhot 010021, PR China. [8]DICP-Surrey Joint Centre for Future Materials, and Advanced Technology Institute, University of Surrey, Guilford, Surrey GU2 7XH, United Kingdom. [9]School of Chemistry and Chemical Engineering, University of Surrey, Guildford, Surrey GU2 7XH, United Kingdom. [10]National Synchrotron Radiation Laboratory, University of Science and Technology of China, Hefei 230029, PR China. [11]These authors contributed equally: Runping Ye, Lixuan Ma, Jianing Mao. ✉e-mail: zhangriguang@tyut.edu.cn; m.duyar@surrey.ac.uk; jiangz@ustc.edu.cn; jian.liu@surrey.ac.uk

However, the poor stability and low methanol selectivity of this catalyst for hydrogenation of pure $CO_2$ have serious impacts on large-scale industrial applications. The methanol selectivity is significantly reduced due to the competing reverse water-gas shift reaction (RWGS). Additionally the high rate of water production poses problems for long term stability[11-15], due to the sintering of Cu nanoparticles and oxide compounds (e.g. ZnO and $ZrO_2$) and the restructuring of their interfaces along with Cu oxidation. The pure Cu-based systems often have low catalytic performance because Cu alone interacts very poorly with $CO_2$ and the apparent activation energy for methanol synthesis is high on Cu(111)[16]. Thus, the multi-metal catalyst system with synergistic effect and abundant interfaces as well as strong metal-support interactions has been extensively developed for addressing this problem[3,17]. In addition to the commercial CuZnAl catalysts for $CO_x$ (x = 1, 2) hydrogenation to methanol[18,19], other multi-metallic catalysts such as CuZnZr[11], CuZnGa[20], CuZnCe[21], CuCeTi[16], and CuZnAlZr have also been reported[22]. Despite the improvements in the catalytic performance of these catalysts, there is still debate regarding the active sites and reaction mechanism of these multi-metal catalyst systems, with the role of oxide and alloy formation being challenging to fully understand[8]. Herein, we combine a concerted experimental and theoretical approach to design an advanced CuZnCe catalyst for methanol synthesis and obtain fundamental understanding as to the nature of active sites and their stability.

Extensive efforts have been undertaken to investigate the above issues with experimental and theoretical approaches, and operando characterization methods. In 2014, Graciani et al. reported that the $CeO_x$/Cu(111) and Cu/$CeO_x$/$TiO_2$(110) interfaces with the combination of metal and oxide sites could have complementary chemical properties to regulate the reaction pathways[16]. In 2017, the active sites over CuZnAl catalyst were further demonstrated to be ZnO/Cu interfacial sites and Zn-Cu bimetallic sites were reconstructed to ZnO/Cu surfaces[23]. However, the active sites are still under debate because Zn-Cu bimetallic alloy sites have also been reported to catalyze the reaction[24-26], especially through the formation of Zn-Cu surface alloy active sites in the presence of CO[27]. Recently, Zabilskiy et al. employed a series of high-pressure operando techniques to further investigate the reaction mechanism on CuZnAl catalyst, and they also observed not only the oxidation of CuZn alloys to ZnO/Cu surfaces, but also the presence of zinc formate as an important reaction intermediate[28]. Interestingly, Beck et al. also observed the existence of zinc formate under a high pressure of 10 bar, however, the zinc formate is difficult to be detected under lower pressure[19]. They also demonstrated that the CuZnAl catalyst composition and morphology were sensitive to the applied pressure and temperature. Therefore, the catalytic active sites and reaction mechanism for methanol synthesis remain under debate without comprehensive agreement.

Metal-organic frameworks (MOFs) with tunable chemical components and tailored structures are an ideal platform to engineer the active sites at the atomic and molecular levels[29,30]. Thus, MOF-based materials can act as templates for the preparation of MOF-derived catalysts at the nanoscale and even at the atomic scale[31,32]. For example, a site-directed reduction strategy was employed to engineer the MOFs/nanoparticle systems with different structures and size-selective properties for ketone hydrogenation[32]. A photoactivated Cu-$CeO_2$ catalyst, which has abundant Cu-$O_V$-Ce active sites derived from the substitution of Cu into Ce-MOF precursor[33], was fabricated through MOFs crystal engineering for the preferential oxidation of CO. Moreover, Yang et al. have demonstrated that the copper-ceria solid solution with enhanced Cu-$O_V$-$Ce_x$ active species could improve the $CO_2$ hydrogenation to methanol[34]. On the basis of these works, we were inspired to engineer the active sites of multi-metal catalysts with more metal incorporated into the support through a MOF crystal engineering strategy. In addition, the hydrogenation of formate species, an important $CO_2$ hydrogenation intermediate, is facilitated by the close contact between the zinc and copper phases[28]. Thus, the MOFs derived catalysts exhibiting intimate contact may present high catalytic activity.

The conventional multi-metal catalysts are usually prepared by impregnation[35], coprecipitation[36], sol-gel[37], or hydrothermal methods[38]. Herein, we synthesized a series of CuZnCe catalysts through MOFs crystal engineering method. As the CuZnCe-MOF was difficult to be synthesized by a one-pot method, the CuZnCe catalysts derived from CuZnCe-MOF were prepared step by step. As a result, we found that the order of introduction of metal during MOFs preparation influences the growth of MOFs, thus influencing the active sites, which would also be influenced by the types of metal species and preparation method. The obtained Ce-CuZn catalyst with abundant Cu/Zn-$O_V$-Ce species exhibited high-performance $CO_2$ hydrogenation to methanol. This was because the introduction of Cu and Zn to $CeO_2$ energetically facilitates $CO_2$ hydrogenation over Ce-CuZn catalyst via formate intermediates, which were observed by in-situ diffuse reflectance infrared Fourier transform spectroscopy (DRIFTS) and the proposed mechanism supported by density functional theory (DFT) calculations. The Cu-$CeO_2$ interactions inhibited the RWGS while further introduction of Zn to decorate Cu-$O_V$-Ce active sites promoted $CO_2$ hydrogenation to methanol via Zn-decorated Cu active sites.

## Results

### Synthetic route

As the ligand of 1,3,5-benzenetricarboxylic acid (1,3,5-BTC) could be coordinated with many metal ions like $Cu^{2+}$, $Zn^{2+}$ and $Ce^{3+}$, we tried to synthesize the CuZnCe-MOF by one-pot method (Route 1 in Supplementary Fig. 1). However, the color was white instead of blue for the dried CuZnCe-MOF sample, indicating that the copper species did not exist in the sample. The X-ray powder diffraction (XRD) patterns of the dried CuZnCe-MOF sample only presented the diffraction peaks of Ce-MOF (namely, Ce-BTC) and the XRD patterns of the calcined CuZnCe sample also only presented the diffraction peaks of $CeO_2$ (Supplementary Fig. 2a, b), proving that the three metals could not grow simultaneously into CuZnCe-MOF sample by one step under these reaction conditions. To solve this problem, we tried to synthesize it by two steps. Firstly, the CuZn-MOF was prepared and then introduced $Ce^{3+}$ into CuZn-MOF to produce CuZn-Ce-MOF (Route 2 in Supplementary Fig. 1). Or the Ce-MOF was prepared firstly and then $Cu^{2+}$/$Zn^{2+}$ into Ce-MOF to produce Ce-CuZn-MOF (Route 3 in Supplementary Fig. 1). The color of the dried CuZn-Ce-MOF and Ce-CuZn-MOF samples were blue and their XRD patterns displayed the diffraction peaks of Cu-MOF. After calcination, the diffraction peaks of CuO and $CeO_2$ could be found, showing the segregation and agglomeration of metal species as well as the decomposition of MOF precursors. The dried Ce-MOF sample lost the solvent molecules at about 150 °C and then lost the 1,3,5-BTC ligand at about 350 °C (Supplementary Fig. 2c). The decomposition temperatures for Ce-CuZn-MOF are lower than Ce-MOF because that Ce-CuZn-MOF was prepared via more steps including solvent washing. Furthermore, the XRD patterns over Ce-CuZn-MOF which originally showed the presence of Ce-MOF, changed upon introduction of $Cu^{2+}$/$Zn^{2+}$ into Ce-MOF (Supplementary Fig. 3), indicating transformation to the Cu-MOF crystal structure. Thus, the crystal structure of Ce-MOF was reconstructed into Cu-MOF for the Ce-CuZn sample, indicating that $Cu^{2+}$/$Zn^{2+}$ were introduced into the frameworks of Ce-MOF and subsequently doped into the structure of $CeO_2$ after calcination (Fig. 1a). To prepare the control sample with the $Cu^{2+}$/$Zn^{2+}$ on the surface of $CeO_2$, the Ce-CuZn-IM was prepared by an impregnation method that involved introduction of $Cu^{2+}$/$Zn^{2+}$ after calcination of Ce-MOF (Route 4 in Supplementary Fig. 1).

### Morphological and textural properties of the catalysts

To further investigate the micro and nanostructure of the Ce-CuZn with optimized catalytic performance, transmission electron

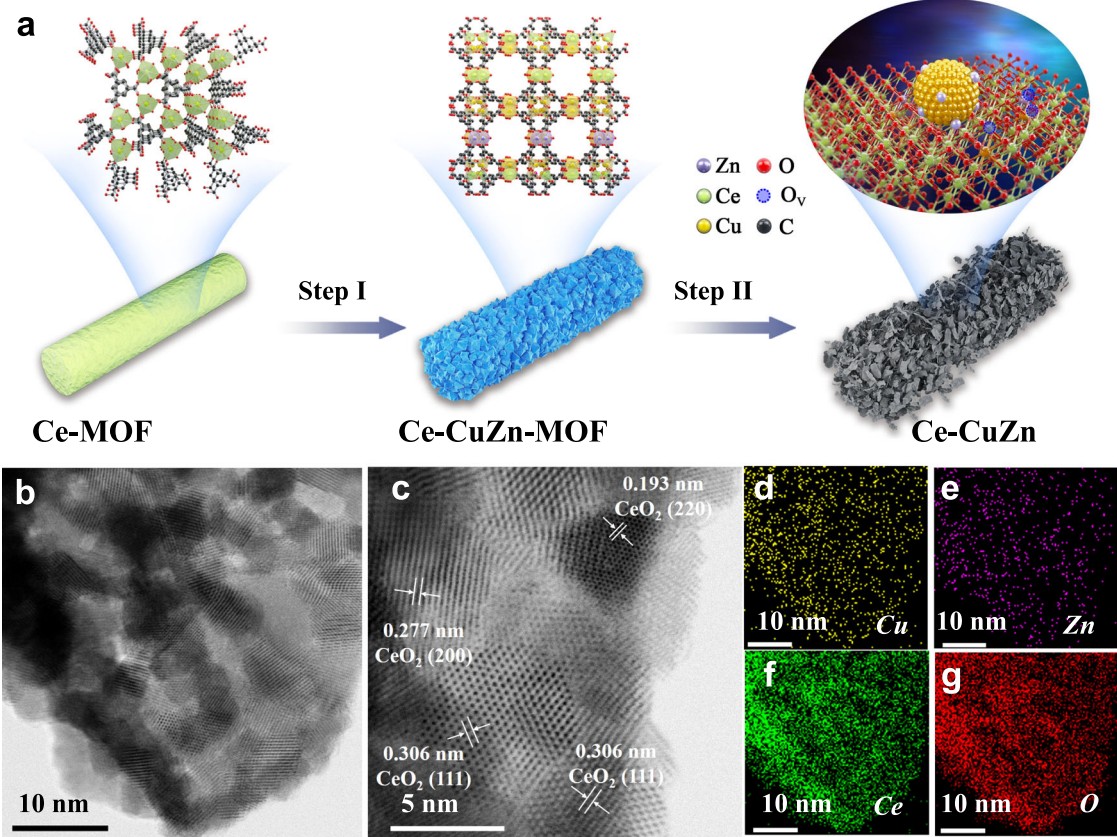

**Fig. 1 | The crystal structure and morphology of Ce-CuZn samples. a** The samples' evolution from Ce-MOF to Ce-CuZn. Setp I: Ultrasonication, and $Cu^{2+}/Zn^{2+}$ ion exchange; Step II: Calcination, reduction, and MOF decomposition to form Cu/Zn-$O_V$-Ce. The purple, green, yellow, red, blue and black balls represent Zn, Ce, Cu, O, $O_V$ and C, respectively. **b–g** The TEM and HRTEM images of reduced Ce-CuZn sample with corresponding elemental mapping.

microscopy (TEM) and scanning electron microscopy (SEM) images of Ce-CuZn sample are presented in Fig. 1 and Supplementary Fig. 4, respectively. The original Ce-MOF presents bundles of smooth nanorods (Supplementary Fig. 4a) while the dried Ce-CuZn-MOF show bundles of coarse nanorods with octahedral structure (Supplementary Fig. 4b, d), which was the typical structure of Cu-MOF as demonstrated by the above XRD results. Upon calcination, the morphology of Ce-MOF also changed from nanorods to the mixture of nanorods, nanosheets, and nanoparticles for the Ce-CuZn-MOF sample (Supplementary Fig. 4c).

The Ce-CuZn nanoparticles were also small and highly dispersed from the TEM image (Fig. 1b). The HRTEM image of Ce-CuZn shows that it could expose different crystal phases of $CeO_2$ (Fig. 1c), such as $CeO_2$ (111), $CeO_2$ (200), and $CeO_2$ (220). The crystal phase of copper could not be observed due to the lighter atomic weight of Cu with respect to Ce and the similar contrast of Cu and $CeO_2$, which was also observed in the other reported $Cu/CeO_2$ catalysts[33,39]. The high-angle annular dark-field scanning transmission electron microscopy (HAADF-STEM) images with the corresponding EDS elemental mapping further show that Ce-CuZn sample exhibited homogeneous of dispersion Cu, Zn, Ce metals (Fig. 1d–g). It should be mentioned that the mean Cu nanoparticle sizes were increased from 5.34 nm over the calcined Ce-CuZn to 8.17 nm over the reduced Ce-CuZn, indicating that the Cu nanoparticles were accompanied by slight migratory agglomeration during reduction process (Supplementary Figs. 5 and 6). However, the solid solution could be kept over the reduced Ce-CuZn sample, which would be demonstrated by the following Raman and electron paramagnetic resonance (EPR) results. For the other two CuZn-Ce and Ce-CuZn-IM samples, the nanorods combined with the nanoparticles could also be observed (Supplementary Figs. 7 and 8). In particular, the partial agglomeration of Cu/Zn nanoparticles were found over the surface of $CeO_2$. Also, the control samples of CuZn-Ce and Ce-CuZn-IM exposed similar crystal phases of $CeO_2$ from the HRTEM images analysis.

The physicochemical properties of CuZnCe series samples are presented in Table 1. The actual Cu, Zn, and Ce loading over the CuZnCe series samples were different. The CuZn-Ce and Ce-CuZn possessed similar copper content of about 52 wt.% but the former had not loaded Zn. However, its precursor of CuZn had 3.55 wt.% of Zn, indicating that the Zn was lost during the second step of adding the Ce. For the Ce-CuZn-IM sample, it has much lower Cu loading (22.36 wt.%) and more Zn loading (11.47 wt.%). This is because the Ce-MOF was calcined to obtain the $CeO_2$ powder and then loaded with Cu and Zn without the centrifugation or washing procedures by the impregnation method. Thus, we regulated the mass of nitrates to synthesize the Ce-CuZn-IM-B sample with a similar metal loading as the Ce-CuZn sample. The CuZnCe series samples present a similar hysteresis loop and specific surface area ($S_{BET}$) of about 41 $m^2$/g, but the Ce-CuZn sample exhibits larger pore size and pore volume (Supplementary Fig. 9 and Table 1). However, the copper surface area and copper dispersion over the Ce-CuZn sample were lower than the control samples (Table 1), indicating that more copper species were doped into the ceria matrix. Although the copper loading over Ce-CuZn-IM was much lower, its surface copper area was similar to that over Ce-CuZn.

### Evolution of crystalline phase and surface properties

The evolution of metal species is illustrated by the XRD results. For the calcination of pure CuZn-MOF, the acute diffraction peaks of CuO are observed (Supplementary Fig. 2b). After introduction of Ce into CuZn-MOF, the diffraction peaks of CuO become broader. All the calcined

**Table 1 | The physicochemical properties of CuZnCe catalysts**

| Catalysts | Cu[a] (wt.%) | Zn[a] (wt.%) | Ce[a] (wt.%) | $S_{BET}$ (m²/g) | Pore size (nm) | Pore volume (cm³/g) | $S_{Cu}$[b] (m²/g) | $D_{Cu}$[b] (%) | $N_{OV}$ (μmol·$g_{cat}$⁻¹)[b] |
|---|---|---|---|---|---|---|---|---|---|
| CuZn | 80.65 | 3.55 | – | 2.9 | 26.9 | 0.01 | 3.7 | 0.7 | – |
| CuZn-Ce | 52.33 | 0.03 | 28.71 | 41.5 | 6.7 | 0.07 | 31.1 | 9.2 | 6.0 |
| Ce-CuZn | 52.82 | 1.12 | 31.74 | 40.2 | 12.1 | 0.12 | 23.1 | 6.7 | 18.1 |
| Ce-CuZn-IM | 22.36 | 11.47 | 48.80 | 41.4 | 7.2 | 0.08 | 22.0 | 15.2 | 2.9 |

[a]Metal loading results from ICP.
[b]Metallic copper surface area ($S_{Cu}$), copper dispersion ($D_{Cu}$), and oxygen vacancies ($N_{OV}$) determined by N₂O titration and H₂ temperature-programmed reduction (H₂-TPR).

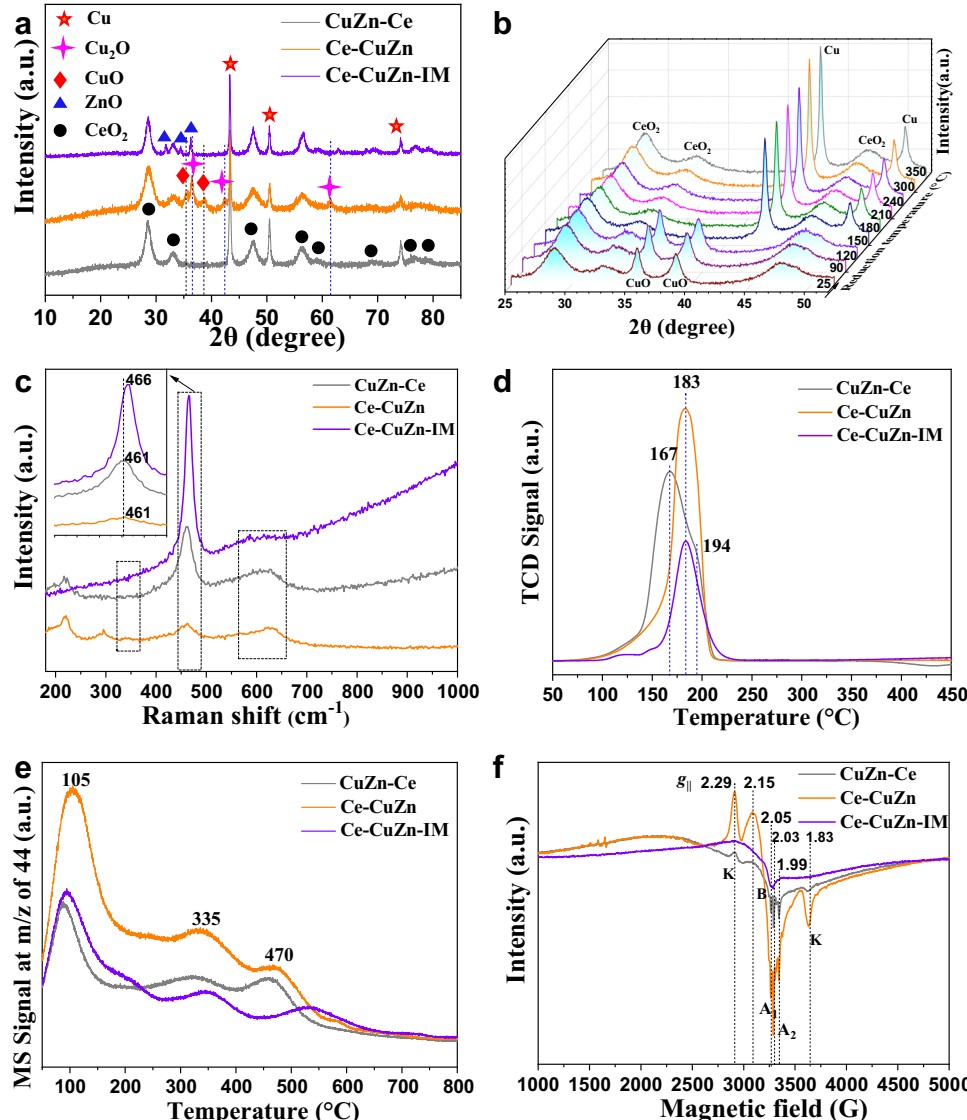

**Fig. 2 | The crystalline phase and surface basicity of CuZnCe catalysts. a** Normal XRD patterns of the reduced samples tested offline. **b** In-situ XRD patterns of Ce-CuZn samples reduced at different temperature under the atmosphere of 40%H₂-N₂. **c** Raman spectra. **d** H₂-TPR curves. **e** CO₂-TPD curves. **f** EPR spectra. K, B, A₁, and A₂ represent the peak signal, where signal K is ascribed to Cu²⁺/Zn²⁺ dimer; signal B is also ascribed to Cu²⁺ ions; signal A₁ has been correlated with isolated Cu²⁺ in octahedral sites in ceria with a tetragonal distortion; signal A₂ has been attributed to isolated Cu²⁺ localized in surface substitutional sites with a square-pyramidal symmetry[56].

CuZnCe series samples show the CuO and CeO₂ species and the Ce-CuZn-IM also presents the ZnO species. After reduction by hydrogen, CuO species can no longer be detected by XRD while the metallic Cu species appear for the CuZn-Ce and Ce-CuZn-IM samples (Fig. 2a). However, the broad diffraction peaks of CuO and sharp diffraction peaks of Cu₂O could be detected over the Ce-CuZn sample. Thus, we further operated the in-situ XRD test under the 40%H₂-N₂ atmosphere for the Ce-CuZn sample (Fig. 2b). The diffraction peaks of CuO and Cu₂O could not be detected in the in-situ XRD results, indicating that the reduced Ce-CuZn sample was very active thus would be facile to be oxidized during offline XRD test. The diffraction peaks over Ce-CuZn sample were also stable during in-situ reaction atmosphere

(Supplementary Fig. 10). This suggests that the copper species were different over CuZnCe series samples. In addition, the Ce-CuZn-IM sample still showed obvious diffraction peaks of ZnO after reduction.

From the Raman spectra (Fig. 2c), the intensity of triply degenerate $F2g$ mode of $CeO_2$ became weaker and there was a distinct blue shift (466 to 461 $cm^{-1}$) from Ce-CuZn-IM to CuZn-Ce and Ce-CuZn samples, suggesting the doping of Cu and Zn to the $CeO_2$ lattice[33]. Simultaneously, an obvious broad band (550 ~ 650 $cm^{-1}$) induced from oxygen vacancies was observed over CuZn-Ce and Ce-CuZn samples, which was attributed to the substitutional incorporation of Cu/Zn ions into the $CeO_2$ lattice[34]. Thus, the Cu/Zn-$O_V$-Ce active sites could be produced from the Cu/Zn substitution into the $CeO_2$ lattice and CuO/ZnO-CeO$_2$ boundary[33], which would also be demonstrated by the following X-ray absorption spectra (XAS) results.

The temperature-programmed reduction (H$_2$-TPR) profiles were fitted with three peaks (Fig. 2d and Supplementary Fig. 11), which were attributed to the reduction of dispersed copper species that weakly interact with $CeO_2$ (peak α), bulk CuO and dispersed copper species that strongly interact with $CeO_2$ (peak β), and Cu/Zn-$O_V$-Ce solid solution (peak γ)[21,40]. It shows that Ce-CuZn and Ce-CuZn-IM samples exhibit higher reduction temperatures than that of CuZn-Ce, which is probably because that the former two samples have higher contents of Zn to decorate the Cu particles[21]. Moreover, the order of peak γ ratio over CuZnCe catalysts is as follows: Ce-CuZn> CuZn-Ce> Ce-CuZn-IM, indicating the existence of many Cu/Zn-$O_V$-Ce species with strong metal-support interaction over the Ce-CuZn sample. The above characterization results indicate that the Ce-CuZn sample exhibits Cu$^0$, Cu$_2$O and Cu$^+$/Zn-$O_V$-Ce species.

The $CO_2$ temperature-programmed desorption ($CO_2$-TPD) experiments were carried out to determine the surface basicity of the catalysts. Three desorption peaks assigned to weak, moderate, and strong basic sites could be observed over CuZnCe series samples (Fig. 2e). Moreover, the Ce-CuZn sample shows a larger $CO_2$ desorption peak and the fitted peak areas of Ce-CuZn is about three times of CuZn-Ce sample, confirming that its surface exhibited weaker basic sites for $CO_2$ adsorption.

Quasi in-situ X-ray photoelectron spectroscopy (XPS) was further performed to analyze the surface species over the reduced CuZnCe series samples. The binding energy of Cu $2p_{3/2}$ over the Ce-CuZn-IM present at 932.8 eV is attributed to Cu$^0$/Cu$^+$ species (Supplementary Fig. 12a). Moreover, the Cu $LMM$ XAES spectra were carried out to determine the specific Cu$^+$/(Cu$^+$+Cu$^0$) ratio as shown in Supplementary Fig. 13a and Supplementary Table 1. The Ce-CuZn sample has a higher Cu$^+$ content (56.5%) and that over Ce-CuZn-IM is not available due to the effect of Zn $2p$. The Ce-CuZn-IM showed weaker peaks of Cu $2p$ but stronger peaks of Zn $2p$, which were almost absent over the CuZn-Ce sample (Supplementary Fig. 12b). This suggests that the surface of Ce-CuZn-IM exhibits many Zn species while CuZn-Ce has no Zn species, which is consistent with the above XRD and ICP results. Moreover, $O1s$ peaks are deconvoluted into three components (Supplementary Fig. 13b), namely lattice oxygen ($O_α$), oxygen vacancies ($O_β$) and surface oxygen ($O_γ$)[41,42]. Similarly, the Ce $3d$ peaks are deconvoluted into four peaks of $3d^{10}4f^1$ Ce$^{3+}$ ($u_0$, $u_1$, $u_0'$, and $u_1'$) and six peaks of $3d^{10}4f^0$ Ce$^{4+}$ ($v_0$, $v_1$, $v_2$, $v_0'$, $v_1'$, and $v_2'$) (Supplementary Fig. 13c). The Ce-CuZn catalyst exhibits a higher ratio (29.9%) of $O_β$ than the other two samples (Supplementary Table 1), indicating that Ce-CuZn catalyst possesses a higher concentration of oxygen vacancies. However, the Ce$^{3+}$/(Ce$^{3+}$+Ce$^{4+}$) ratio (34.8%) of Ce-CuZn is slightly lower than that over CuZn-Ce (36.4%), but higher than that over Ce-CuZn-IM (27.4%, Supplementary Table 1). This is because that the catalyst has two types of oxygen vacancies: I) generation from the reduction of Ce$^{4+}$ to Ce$^{3+}$, and II) the replacement of Ce$^{4+}$ by Cu/Zn ions, resulting in the formation of oxygen vacancies[43]. Thus, the Ce-CuZn catalyst with slightly lower Ce$^{3+}$/(Ce$^{3+}$+Ce$^{4+}$) ratio could still exhibit higher $O_β$/($O_α$ + $O_β$ + $O_γ$) ratio due to the substitutional incorporation of Cu/Zn ions into the $CeO_2$

lattice. Therefore, the oxygen vacancies results at 531.2 eV in O $1s$ spectra and the Ce$^{3+}$/(Ce$^{3+}$+Ce$^{4+}$) ratio in Ce $3d$ spectra are consistent with the Raman results.

Moreover, EPR and chemisorption measurements were carried out to confirm the oxygen vacancies. The EPR spectrum of Ce-CuZn sample presents obvious oxygen vacancies peak at $g_{||}$ value of 1.99, which is much different with the other two samples (Fig. 2f)[43]. Moreover, the type K peaks at $g_{||}$ values of 2.29 and 1.83 are ascribed to Cu$^{2+}$/Zn$^{2+}$ dimer, which could be observed when two neighboring Ce$^{4+}$ ions with short separation distance are substituted by Cu$^{2+}$/Zn$^{2+}$ ions[44]. Thus, the appearance of K signals suggests that Cu/Zn-$O_V$-Ce solid solution is indeed generated in Ce-CuZn sample. In addition, the chemisorption measurement to determine the amount of oxygen vacancies ($N_{OV}$) was developed by Zhu et al.[40], and the results are summarized in Table 1. It presents that the Ce-CuZn sample exhibits 18.1 μmol·g$_{cat}^{-1}$ of oxygen vacancies, which is still much higher than the other two samples (2.9 ~ 6.0 μmol·g$_{cat}^{-1}$). These results are consistent with the above quasi in-situ XPS and the EPR results.

To further obtain the quantitative information of electron and coordination environment, XAS analysis was conducted. The fingerprint effect of X-ray absorption near edge structure (XANES) ascertains the valence state of the absorption atom. Figure 3a shows the spectra of normalized Cu K-edge XANES and corresponding standard samples. As demonstrated by the XANES of Ce-CuZn and Ce-CuZn-IM of Cu K-edge, the absorption edge slightly shifts to higher values compare with Cu foil, implying the presence of oxidized copper in Ce-CuZn and Ce-CuZn-IM. Meanwhile, the oxidation of Ce-CuZn-IM exhibited a slightly high oxidation state. We then resorted to extended X-ray absorption fine structure (EXAFS) spectra to investigate the local structure. The Fourier transform of Cu K-edge EXAFS result in Fig. 3b reveals that the Ce-CuZn and Ce-CuZn-IM demonstrates a weak path at around 1.4 Å and a predominant peak at 2.2 Å, corresponding to Cu-O and Cu-Cu scattering path, respectively. Least-squares EXAFS fitting analysis was further adopted (Supplementary Fig. 14 and Supplementary Table 2), with best fitting analysis of Ce-CuZn showing the dominance of Cu-Cu bond with coordination number of 10.0 and a low Cu-O contribution was determined as 0.2. For Zn, the normalized XANES showed in Fig. 3c, and the spectrum of Ce-CuZn is in the same position with ZnO; for comparison, the position of Ce-CuZn-IM is between Zn foil and ZnO, implying that the valence state of Zn in Ce-CuZn is Zn$^{2+}$ and in Ce-CuZn-IM is the mixture of Zn$^0$ and Zn$^{2+}$, and the specific valence state is 1.6 via linear combination fitting (LCF). The evolution of coordination configuration of Ce-CuZn and Ce-CuZn-IM was identified by EXAFS (Fig. 3d–f). The fitting results of coordination environment of Zn K-edge spectra show more differences (Supplementary Fig. 14 and Supplementary Table 3). The best EXAFS fitting of Ce-CuZn gives 4.5 Zn-O bonds ($R = 1.96$Å) and 11.5 Zn-Zn bonds ($R = 3.23$Å); and the Ce-CuZn-IM exhibits 3.3 Zn-O bonds ($R = 1.98$Å) and 12.0 Zn-Zn bonds ($R = 3.24$Å). Since the higher oxidation state of Zn in Ce-CuZn, and almost no CuZn alloy generated in the catalyst, we speculate that some Cu and Zn were incorporated into $CeO_2$ lattice to form Cu/Zn-$O_V$-Ce species while the other Cu and Zn on the catalyst surface combined with $CeO_2$ to form Cu/Zn-ceria interfaces. While the Zn species over the Ce-CuZn-IM samples are mainly ZnO, and also a part of CuZn alloy and Cu/Zn-$O_V$-Ce species.

## Catalytic activity and stability

The catalytic performance of $CO_2$ hydrogenation to methanol on CuZnCe series samples was evaluated in a fix-bed reactor (Fig. 4a) and the results are shown in Fig. 4 and Supplementary Table 4. The Ce-CuZn displayed the highest Con.$_{CO2}$ and space-time yield (STY) of methanol (140.6 g·g$_{cat}^{-1}$·h$^{-1}$) under identical reaction conditions ($P = 2.0$ MPa, $T = 280$ °C, $GHSV = 10,000$ mL·g$_{cat}^{-1}$·h$^{-1}$) among CuZnCe series samples (Fig. 4b). While the Ce-Zn, CuZn and Ce-CuZn-IM samples exhibited higher methanol selectivity but much lower

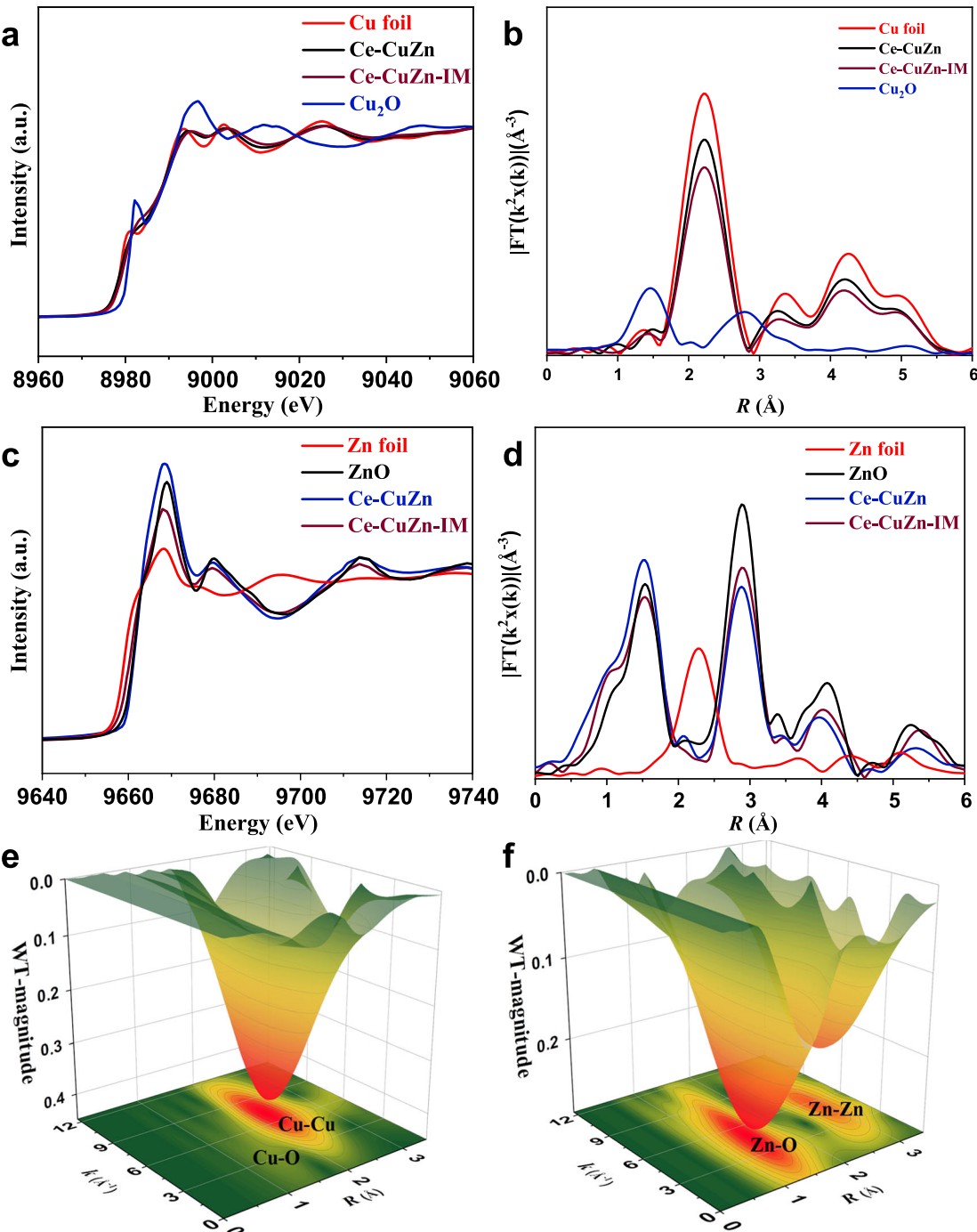

**Fig. 3 | XAS spectra of reduced Ce-CuZn and Ce-CuZn-IM samples. a** Cu K-edge XANES and corresponding standard samples. **b** Fourier-transformed $k^2$-weight EXAFS of Cu K-edge. **c** Zn K-edge XANES and corresponding standard samples. **d** Fourier-transformed $k^2$-weight EXAFS of Zn K-edge. **e, f** The WT spectroscopy of Ce-CuZn sample.

$CO_2$ conversion, as a result with lower STY of methanol ($5.5 \sim 26.8$ g·kg$_{cat}^{-1}$·h$^{-1}$, Supplementary Table 4). When compared at similar $CO_2$ conversion, the Ce-CuZn still presented the highest methanol selectivity and STY of methanol (Supplementary Fig. 15). We further calculated the TOF values of CuZnCe series catalysts (Supplementary Table 5), and the results show that the TOF value ($19.0$ h$^{-1}$) of Ce-CuZn is $5 \sim 8$ times of the other CuZnCe series catalysts ($2.3 \sim 3.4$ h$^{-1}$).

To assess the effect on performance of the order in which the ions are introduced, more control samples have been prepared and evaluated, as illustrated in Supplementary Table 4. The pure $CeO_2$ support and the binary system of CuZn, Ce-Cu, and Ce-Zn catalysts have presented poor catalytic performance. Thus, it is necessary to prepare the ternary system. For the order of introducing the Cu, Zn, and Ce elements, they were firstly introduced together, but the CuZnCe sample exhibited only 1.4% of Con.$_{CO2}$ at 280 °C. The CuZnCe sample only grew Ce-MOF with very low CuZn and showed poor performance. Thus, the three metals could not be added together bringing about the question of which element should be introduced first. When the Cu was introduced first, the CuZn-Ce sample with low content of Zn (0.03%, Table 1) presented low performance as the Zn was lost during the second step of preparation. When the Zn was introduced first, the Zn-CuCe sample could not be prepared because that Zn-MOF was not generated under the similar conditions. When the Ce was introduced

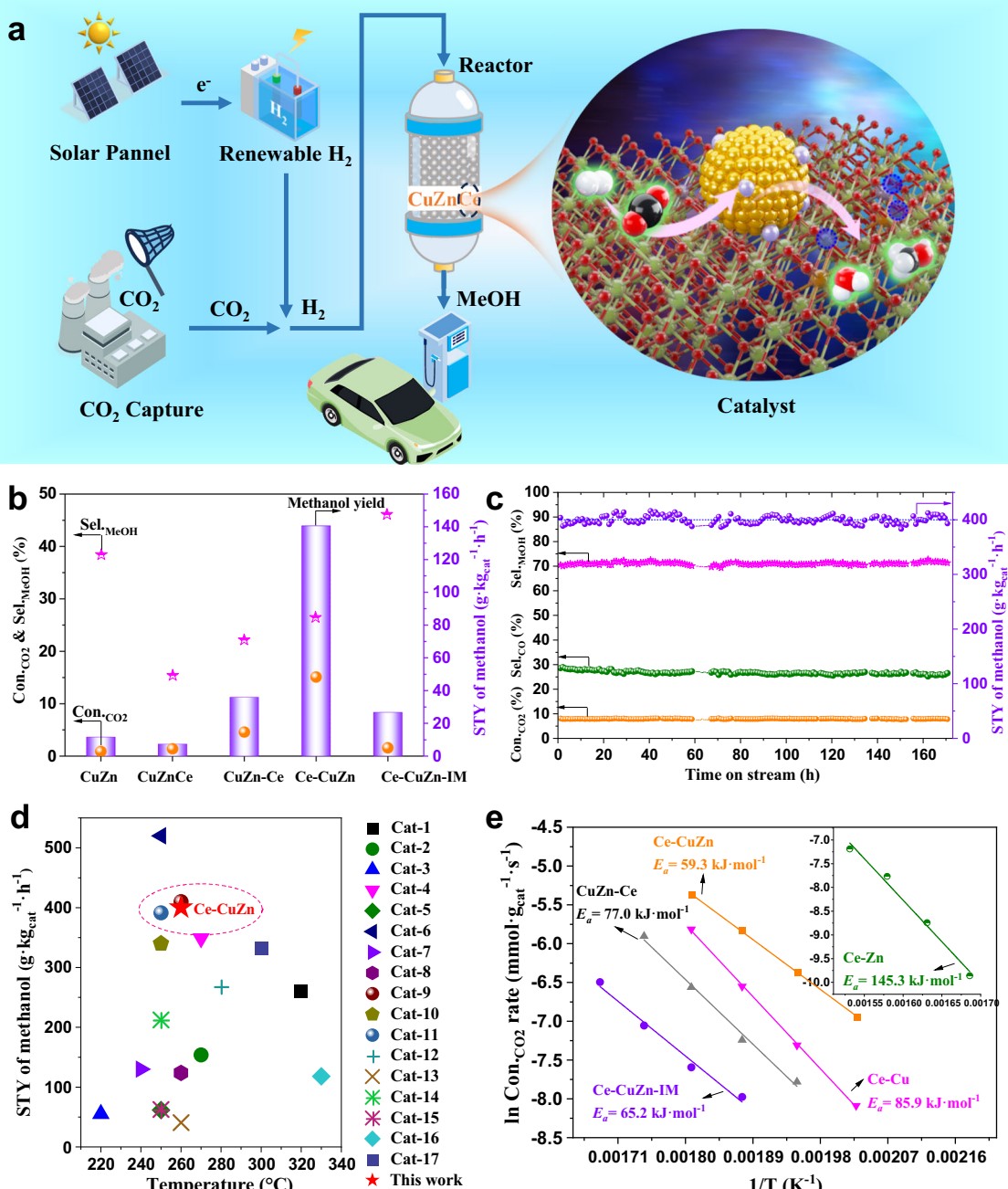

**Fig. 4 | Catalytic performance for CO$_2$ hydrogenation over CuZnCe catalysts.**
**a** Schematic illustration of the thermal CO$_2$ hydrogenation to methanol. The purple, green, yellow, red, blue, black and white balls represent Zn, Ce, Cu, O, O$_V$, C and H, respectively. **b** Catalytic performance under the identical reaction conditions. **c** Catalytic stability of Ce-CuZn catalyst. **d** Comparison of the Ce-CuZn sample with the reported methanol synthesis catalysts under the reaction conditions of 2–5 MPa, 1500–48,000 mL·g$_{cat}^{-1}$·h$^{-1}$, and 220–330 °C (Supplementary Table 6). The reference data is cited below: Cu/SiO$_2$-AE (Cat-1)[57], Cu/ZnO@m-SiO$_2$ (Cat-2)[58], Cu/ZnO/SiO$_2$ (Cat-3)[59], CuZnGa/SiO$_2$ (Cat-4)[60], Cu/Ga$_2$O$_3$/ZrO$_2$ (Cat-5)[61],

Cu/Zn/Al/Y (Cat-6)[62], CuZnAl-C-1.25 (Cat-7)[63], Cu/ZnO/Al$_2$O$_3$-1 (Cat-8)[64], Cu/ZnO/Al$_2$O$_3$-2 (Cat-9)[65], Cu/ZnO/Al$_2$O$_3$-3 (Cat-10)[62], Cu/ZnO/Al$_2$O$_3$-4 (Cat-11)[66], Cu/CeO$_2$ (Cat-12)[52], Cu-Pd/CeO$_2$ (Cat-13)[67], CuZn@UiO-bpy (Cat-14)[66], CuZn-BTC (Cat-15)[68], In$_2$O$_3$ (Cat-16)[69], and h-In$_2$O$_3$-R (Cat-17)[70]. **e** Apparent activation energy ($E_a$) determined by Arrhenius plots based on CO$_2$ hydrogenation. Reaction conditions: **b** $P$ = 2.0 MPa, $T$ = 280 °C, $GHSV$ = 10,000 mL·g$_{cat}^{-1}$·h$^{-1}$, H$_2$: CO$_2$: N$_2$ = 72: 24: 1; **c** $GHSV$ = 20,000 mL·g$_{cat}^{-1}$·h$^{-1}$, $P$ = 2.8 MPa, $T$ = 260 °C, H$_2$: CO$_2$: N$_2$ = 72: 24: 1; **e** $P$ = 2.0 MPa, $GHSV$ = 20,000 mL·g$_{cat}^{-1}$·h$^{-1}$ for Ce-CuZn and 10,000 mL·g$_{cat}^{-1}$·h$^{-1}$ for the others, H$_2$: CO$_2$: N$_2$ = 72: 24: 1. $E_a$ is the apparent activation energy.

first, the CeCu-Zn with low Cu content and CeZn-Cu samples with low Zn content have bad performance because Ce-MOF could not be grown together with Cu/Zn-MOF. However, the Ce-CuZn sample could grow Ce-MOF and Cu/Zn-MOF well with two main steps, and thus it had high Cu/Zn-O$_V$-Ce species and exhibited the best performance. When the Zn content was increased, the Ce$_2$-CuZn$_2$ and Ce$_1$-CuZn$_4$ samples would decrease CO$_2$ conversion although the methanol yield was slightly increased. In addition, we also investigated more complicated

preparation procedures with three main steps. It was shown that the Ce-Cu-Zn and Ce-Zn-Cu samples also have good performance, but the preparation procedures are more complicated. Furthermore, although the Sel.$_{MeOH}$ and STY$_{MeOH}$ slightly increased over Ce$_2$-CuZn$_2$, Ce$_1$-CuZn$_4$ and Ce-Cu-Zn samples, they were increased at the expense of CO$_2$ conversion compared with Ce-CuZn sample. When the Con.$_{CO2}$ over Ce-CuZn sample was also near 10%, the Sel.$_{MeOH}$ and STY$_{MeOH}$ were increased to 45.5% and 154.0 g·kg$_{cat}^{-1}$·h$^{-1}$, respectively. Therefore,

the optimized Ce-CuZn catalyst has the appropriate metal elements and suitable introduction order to form abundant Cu/Zn-O$_V$-Ce species, thus it presents the best catalytic performance for methanol synthesis compared with the control catalysts.

As the Ce-CuZn-IM sample possessed different Cu and Zn contents compared with Ce-CuZn, we also tested the performance of Ce-CuZn-IM-B sample that possessed similar Cu and Zn contents compared with Ce-CuZn sample. As a result, the Ce-CuZn-IM-B samples still showed much lower $CO_2$ conversion and STY of methanol compared with Ce-CuZn sample (Supplementary Table 4). Furthermore, the actual surface Cu contents of Ce-CuZn and Ce-CuZn-IM were similar (Supplementary Table 1), but their performance was very different. Thus, this indicates that the state of Zn promoter and intimate contact of active species in Cu/Zn-O$_V$-Ce, influenced by the preparation method and the order or method of metal introduction, play an essential role for direct $CO_2$ hydrogenation to methanol.

Supplementary Fig. 16 shows that the optimized reaction temperature for the STY of methanol was 260 °C over the Ce-CuZn catalyst. Thus, the long-term stability test was further carried out at 260 °C. The average Con.$_{CO2}$, Sel.$_{MeOH}$, Sel.$_{CO}$, and STY of methanol over Ce-CuZn sample during time on stream of 170 h were 8.0%, 71.1%, 26.7%, and 400.3 g·kg$_{cat}^{-1}$·h$^{-1}$, respectively (Fig. 4c), which were stable without obvious decrease. Compared to the some technical CuZnAl catalysts, the Ce-CuZn have presented lower CO selectivity (Supplementary Table 6). Moreover, the STY of methanol on CuZnAl catalyst was decreased about 17% from 180 to 150 g·kg$_{cat}^{-1}$·h$^{-1}$ after about 150 h[15]. This indicates that the Ce-CuZn sample here is a robust catalyst with excellent stability, outperforming the commercial CuZnAl. Compared with the state-of-the-art catalysts reported in the literature, the Ce-CuZn sample also

exhibits comparable STY of methanol under similar reaction conditions (Fig. 4d and Supplementary Table 6).

As the Ce-CuZn catalyst possessed superior $CO_2$ hydrogenation performance, we further operated the CO hydrogenation and apparent activation energy ($E_a$) tests. As illustrated in Supplementary Fig. 17, the Ce-CuZn catalyst also presented high methanol selectivity (81.0%) and yield (225.6 g·kg$_{cat}^{-1}$·h$^{-1}$) at 300 °C during CO hydrogenation, suggesting that the CO produced during $CO_2$ hydrogenation via the RWGS reaction could be recycled and hydrogenated on Ce-CuZn catalyst again. Furthermore, the apparent activation energy on Ce-CuZn (59.3 kJ·mol$^{-1}$) is much lower than the other CuZnCe series catalysts (Fig. 4e), especially lower than the Ce-Zn and Ce-Cu samples, indicating that Cu and Zn simultaneously doping into CeO$_2$ via the MOFs crystal engineering method can significantly improve the activation of the reactant molecules.

### Reaction mechanism and DFT calculations

The above preliminary characterization results show that the Cu/Zn-O$_V$-Ce active sites could be produced from the Cu/Zn substitution into the CeO$_2$ lattice and CuO/ZnO-CeO$_2$ boundary. We further performed in-situ CO-DRIFTS to investigate the surface interactions with the CO probe molecule. The peaks at 2173 cm$^{-1}$ are attributed to CO adsorbed on the CeO$_2$ surface[45], thus it indicates that the Ce-Zn and Ce-Cu samples exhibit some CeO$_2$ surfaces. The main peaks at 2120–2106 cm$^{-1}$ are resulted from CO adsorbed on Cu$^+$ species while the CO adsorbed on Cu$^0$ species were not obvious at around 2058 cm$^{-1}$ (Fig. 5a), which was probably because the in-situ CO-DRIFTS tests were operated at room temperature and the CO adsorption on Cu$^0$ species was weak[45]. It shows that the Ce-CuZn-IM exhibits the largest CO adsorption peak while the Ce-Zn almost has no peak, indicating that

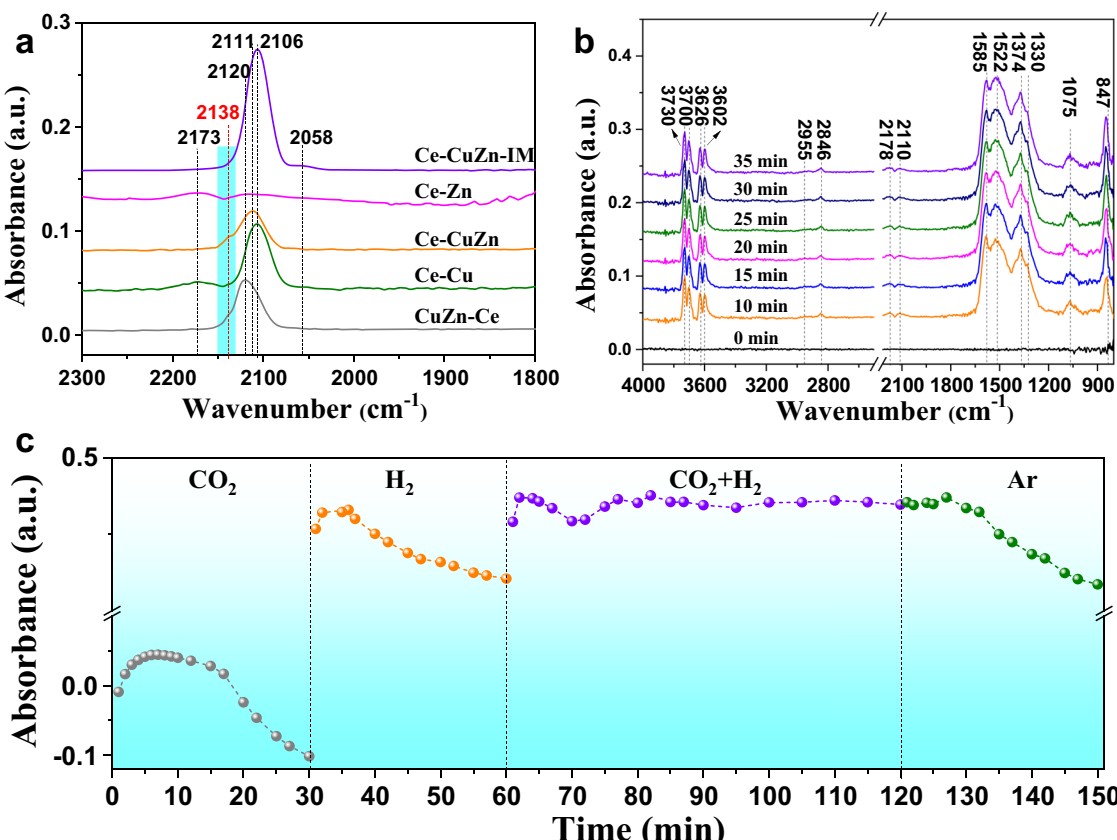

**Fig. 5 | The investigation of potential reaction mechanism on Ce-CuZn catalyst. a** CO-DRIFTS results of CuZnCe after purged by He for 20 min at 30 °C. **b** The result of in-situ DRIFTS for $CO_2$ hydrogenation at 260 °C. **c** Transient in-situ DRIFTS

experiments on Ce-CuZn sample, and the peak intensity of HCOO$^*$ species at 1585 cm$^{-1}$ as a function of time when the reaction atmospheres were switched from $CO_2$ to $H_2$, $CO_2$ + $H_2$, and Ar.

the surface of Ce-CuZn-IM possesses abundant copper species. In addition, the Ce-CuZn and CuZn-Ce samples present an obvious shoulder peak at 2138 cm$^{-1}$, which belongs to single-site Cu$^+$ ions located in constrained environments (Fig. 5a)[46–48]. Therefore, the in-situ CO-DRIFTS confirmed the existence of Cu$^+$ species on the CuZnCe catalysts.

To investigate the potential reaction mechanism, in-situ DRIFTS experiments of $CO_2$ hydrogenation were carried out on Ce-CuZn and Ce-CuZn-IM catalysts. As shown in Supplementary Fig. 18 and Fig. 5b, the main adsorption peaks over Ce-CuZn-IM and Ce-CuZn catalysts are similar except the peaks at 3730–3602 cm$^{-1}$, which are attributed to the terminal, bridged, and triply bridged hydroxyls as well as the Ce$^{3+}$-OH and bicarbonate (Fig. 5b)[49]. Both Ce-CuZn and Ce-CuZn-IM catalysts have presented strong peaks of HCOO* species at 2846, 1585, and 1374 cm$^{-1}$, which were corresponding to the CH stretch mode, symmetric and asymmetric OCO stretching modes[12,34]. However, the gas phase CO* peak (2178 cm$^{-1}$) and CO adsorbed peak (2110 cm$^{-1}$) on Cu$^+$ species were weak[39,50,51]. The other surface species such as monodentate carbonates (1522 cm$^{-1}$)[52], polydentate carbonates (1330, 847 cm$^{-1}$)[39], and methoxy species (2955, 1075 cm$^{-1}$) could also be observed from the in-situ DRIFTS spectra[12,53]. Thus, the in-situ DRIFTS experiments demonstrate that the Ce-CuZn and Ce-CuZn-IM catalysts have the same main reaction intermediates of formate species.

Moreover, the transient in-situ DRIFTS experiments were performed to investigate the role of formate in the mechanism. The Ce-CuZn catalyst was firstly exposed to pure $CO_2$ then the system was switched to other reaction atmospheres (H$_2$, $CO_2$ + H$_2$, Ar), resulting in significant change of the surface species, as shown in Fig. 5c and Supplementary Fig. 19. Firstly, when the $CO_2$ gas was injected to the system, different types of hydroxyl groups at around 3730 ~ 3602 cm$^{-1}$ appeared and increased with the time. Some other carbonates (1522, 1330 cm$^{-1}$) and formate (1585, 1374 cm$^{-1}$) peaks are also shown in Supplementary Fig. 19a. When the gas was switched to H$_2$, the above peaks became weak and peaks of hydroxyl groups disappeared (Supplementary Fig. 19b). In particular, the intensity of formate at 1585 cm$^{-1}$ increased firstly and then decreased slightly when the system was dosed $CO_2$ (Fig. 5c), which was because that $CO_2$ firstly reacted with hydrogen available on the catalyst surface to generate formate but then decreased due to conversion to methoxy and lack of hydrogen. Thus, when the hydrogen was injected into the system, the intensity of formate increased quickly and then decreased with time.

Secondly, upon switching the reaction atmosphere from hydrogen to $CO_2$ + H$_2$, the OH* peaks became positive but then became negative after 120 min (Supplementary Fig. 19c), which was much different compared with the pure $CO_2$ atmosphere. It was possible that the surface hydroxyl groups reacted with $CO_2$ and the bicarbonates were converted to methanol. The formate peaks also decreased first and then increased to a stable state during reaction. Finally, the gas flow was switched to Ar, and the OH* peaks disappeared and other species peaks became weaker and weaker. However, the formate and carbonates could still be observed after purging for 30 min (Supplementary Fig. 19d), indicating that these surface species were stable. In addition, the CO* peaks at 2178 and 2110 cm$^{-1}$ could not be observed during the transient in-situ DRIFTS experiments, thus these CO* species were regarded mainly as spectators during the reaction. Therefore, the surface species of formate, carbonates, bicarbonate, and methoxy are proposed as the main reaction intermediates in the mechanism of $CO_2$ hydrogenation to methanol.

DFT calculations were further implemented to investigate the adsorption and activation of H$_2$ and $CO_2$, as well as the potential reaction mechanism over CuZnCe catalysts. Combined with CO-DRIFTS and XPS results, it can be concluded that the Ce-CuZn sample exhibits both the Cu$^0$ and Cu$^+$ species, in which the content of Cu$^+$ species is up to 56.5%, indicating that abundant surface Cu$^+$ species are responsible for the formation of Cu/Zn-O$_V$-Ce active sites on the Ce-

CuZn sample. Meanwhile, the EXAFS result (Fig. 3) showed that the Cu-O and Cu-Cu scattering path existed in the Ce-CuZn sample, thus, two Cu atoms are considered to replace the Ce atoms to reflect the Cu-Cu and Cu-O coordination structures. Firstly, we adopt the dominantly exposed (111) facet and further construct three models to stimulate CeO$_2$ sample with oxygen vacancy, Cu doped Ce-Cu sample and Cu/Zn co-doped Ce-CuZn sample (Supplementary Fig. 20), which are named as the O$_V$-CeO$_2$, Cu$^+$-CeO$_{2-x}$ and Zn/Cu$^+$-CeO$_{2-x}$, respectively. Given that Cu$^+$ acts as the main species in the Ce-CuZn sample, the differential charge density and Bader charge are employed to characterize the electronic effect of dopant Cu and Zn atoms.

As shown in Fig. 6a, the Cu ions have the charge of −0.49/−0.55 $e$ over Cu$^+$-CeO$_{2-x}$ and those are close to bulk Cu$_2$O (−0.53 $e$), indicating that the Cu ions are correctly displayed to be the +1 valence state. Over Zn/Cu$^+$-CeO$_{2-x}$, the Cu ions have the similar charge of −0.44/−0.59 $e$ with that over Cu$^+$-CeO$_{2-x}$, while the Zn ion has much higher charge of −0.68 $e$ than Cu$^+$ ions, indicating that the more electrons are transferred from the neighboring vacancies to Zn ion compared with Cu ions and thus charge accumulation around the Zn-O$_V$ sites. Moreover, the previous work reported the promotion effect of Zn species in $CO_2$ hydrogenation to methanol, for example, Liu et al. have demonstrated that defective ZnO$_{1-x}$/Cu interfaces present superior activity toward methanol synthesis[54]. Thus, the enriched charge around the Zn-O$_V$ sites makes it possible to act as the active regions to facilitate the adsorption and activation of $CO_2$ and H$_2$, as well as $CO_2$ hydrogenation.

H$_2$ adsorption and activation over O$_V$-CeO$_2$, Cu$^+$-CeO$_{2-x}$ and Zn/Cu$^+$-CeO$_{2-x}$ catalysts are firstly examined, and the corresponding configurations of H$_2$ molecular and dissociative adsorption are presented in Supplementary Fig. 21 and Supplementary Table 7. Over O$_V$-CeO$_2$, H$_2$ dominantly exists in the form of molecular adsorption. Over Cu$^+$-CeO$_{2-x}$, H$_2$ is molecular adsorption at the Cu$^+$ site, while the adjacent oxygen site of Cu$^+$ site promotes the spontaneous homolytic dissociation of H$_2$. Over Zn/Cu$^+$-CeO$_{2-x}$, Cu$^+$ site is still dominantly responsible for molecular adsorption H$_2$, however, the Zn-O$_V$ active regions induced the polarization of Zn−O bonds, which favor the heterolytic dissociation of H$_2$.

The dissociation of molecular adsorption H$_2$ over O$_V$-CeO$_2$, Cu$^+$-CeO$_{2-x}$ and Zn/Cu$^+$-CeO$_{2-x}$ catalysts are further analyzed, as presented in Fig. 6b and Supplementary Fig. 22. It can be seen that the dissociation of H$_2$ is exothermic over these three surfaces, correspondingly, the activity of H$_2$ dissociation follows the order of O$_V$-CeO$_2$ (133.1 kJ·mol$^{-1}$) < Cu$^+$-CeO$_{2-x}$ (109.1 kJ·mol$^{-1}$) < Zn/Cu$^+$-CeO$_{2-x}$ (84.8 kJ·mol$^{-1}$), namely, compared to CeO$_2$ catalyst, Cu-doped CeO$_2$ favor the dissociation of H$_2$, and the addition of Zn into Cu$^+$-CeO$_{2-x}$ further promotes H$_2$ dissociation.

Also, as presented in Supplementary Fig. 23 and Supplementary Table 7, both the linear and bent adsorption configurations of $CO_2$ over O$_V$-CeO$_2$, Cu$^+$-CeO$_{2-x}$ and Zn/Cu$^+$-CeO$_{2-x}$ catalysts are observed. $CO_2$ dominantly exists in the form of carbonate with the interaction of a surface oxygen atom, which is more energetically favorable compared to the linear $CO_2$ adsorption. Meanwhile, $CO_2$ adsorption energies in the form of carbonate follow the order of O$_V$-CeO$_2$ (−30.6 kJ·mol$^{-1}$) < Cu$^+$-CeO$_{2-x}$ (−48.2 kJ·mol$^{-1}$) < Zn/Cu$^+$-CeO$_{2-x}$ (−58.5 kJ·mol$^{-1}$), suggesting that the stronger $CO_2$ adsorption induced by the doping with Cu and Cu/Zn is attributed to the formation of Cu$^+$ and Zn-O$_V$ active sites.

For CO adsorption, as presented in Supplementary Fig. 24 and Supplementary Table 7, CO prefers to adsorb at the O$_V$ site on the O$_V$-CeO$_2$, while the most favorable adsorption site over Cu$^+$-CeO$_{2-x}$ and Zn/Cu$^+$-CeO$_{2-x}$ catalysts is Cu$^+$ site, which is consistent with the CO-DRIFTS results. Moreover, CO adsorption energies follow the order of Zn/Cu$^+$-CeO$_{2-x}$ (−87.5 kJ·mol$^{-1}$)>Cu$^+$-CeO$_{2-x}$ (−61.3 kJ·mol$^{-1}$) > O$_V$-CeO$_2$ (−36.7 kJ·mol$^{-1}$), and the C−O bond lengths also follow the same order of Zn/Cu$^+$-CeO$_{2-x}$ (1.171 Å) >Cu$^+$-CeO$_{2-x}$ (1.167 Å)>O$_V$-CeO$_2$ (1.141 Å), indicating that both the

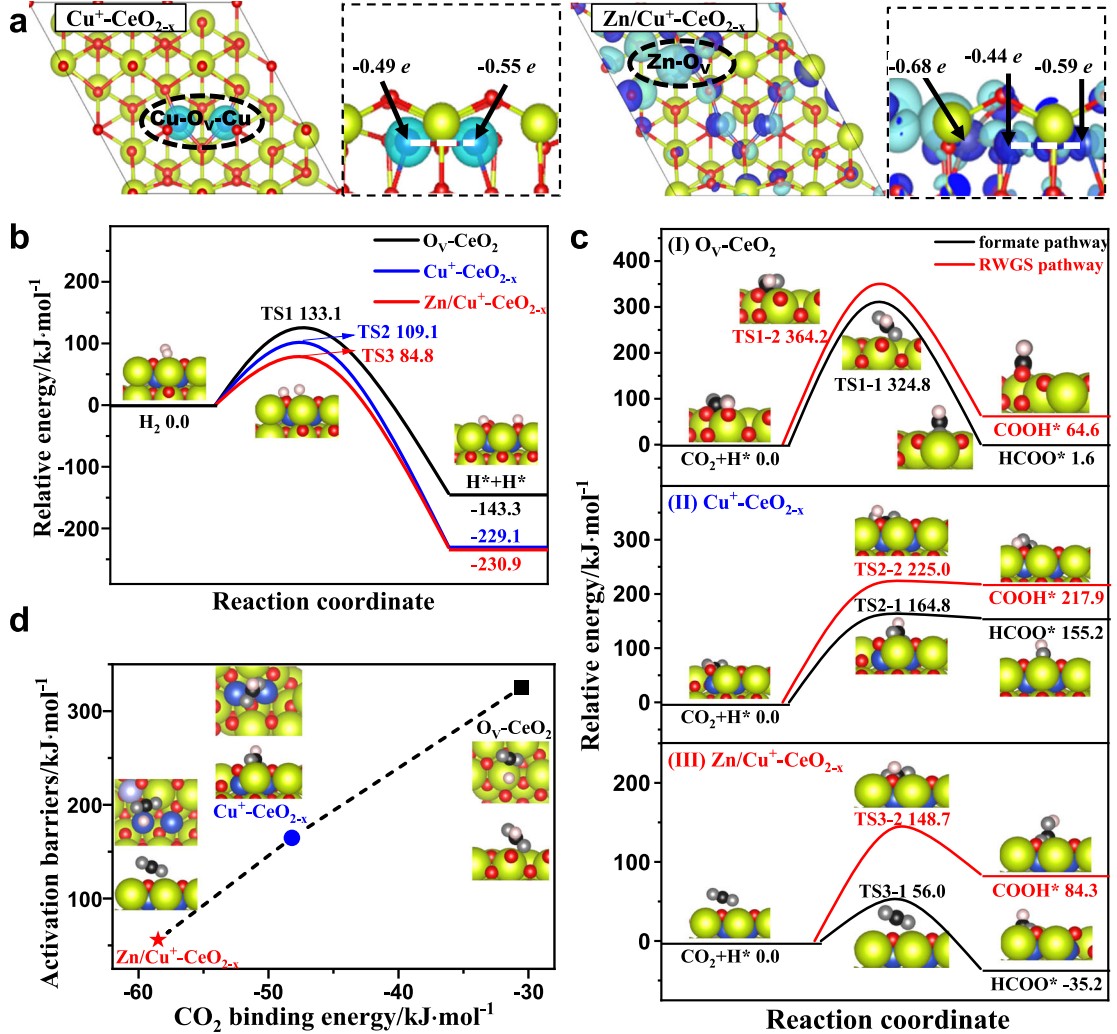

**Fig. 6 | DFT calculation results over the $O_V$-CeO$_2$, Cu$^+$-CeO$_{2-x}$ and Zn/Cu$^+$-CeO$_{2-x}$ catalysts to investigate the reaction mechanism. a** Differential charge density ($\Delta\rho$) of Cu$^+$-CeO$_{2-x}$ and Zn/Cu$^+$-CeO$_{2-x}$ catalysts, the dark blue and light blue regions represent the charge accumulation and depletions. The potential energy profiles and the corresponding structures involved in **b** the dissociation of molecular adsorption H$_2$ and **c** CO$_2$ activation. **d** The relationship of CO$_2$ binding energy with the activation barrier of CO$_2$ hydrogenation to HCOO*. Green: Ce, purple: Zn, blue: Cu, black: C, white: H; red represents the surface O and gray represents O in adsorbed molecules. RWGS reverse water-gas shift reaction, TS transition state.

doping of Cu and Cu/Zn promote the adsorption of CO molecule at the Cu$^+$ site over Cu$^+$-CeO$_{2-x}$ and Zn/Cu$^+$-CeO$_{2-x}$ catalysts.

Furthermore, previous studies regarded the formate (HCOO*) and carboxylate (COOH*) species as the key intermediates in the formate and RWGS pathways for CO$_2$ conversion[11,41]. Thus, both HCOO* and COOH* intermediates are considered for CO$_2$ activation over $O_V$-CeO$_2$, Cu$^+$-CeO$_{2-x}$ and Zn/Cu$^+$-CeO$_{2-x}$ catalysts, as shown in Supplementary Table 8. The potential energy profiles and the corresponding structures are presented in Fig. 6c and Supplementary Fig. 25. Over $O_V$-CeO$_2$, CO$_2$ hydrogenation to HCOO* has the activation barrier and reaction energy of 324.8 and 1.6 kJ·mol$^{-1}$, which is kinetically and thermodynamically superior to COOH* formation (364.2 and 64.6 kJ·mol$^{-1}$). The same also occurs over Cu$^+$-CeO$_{2-x}$ and Zn/Cu$^+$-CeO$_{2-x}$. The formate pathway for CO$_2$ activation is dominant over these three catalysts with the activity order of $O_V$-CeO$_2$ (324.8 kJ·mol$^{-1}$) < Cu$^+$-CeO$_{2-x}$ (164.8 kJ·mol$^{-1}$) < Zn/Cu$^+$-CeO$_{2-x}$ (56.0 kJ·mol$^{-1}$).

Overall, DFT calculations unraveled that CO$_2$ hydrogenation to produce HCOO* is more favorable both in kinetics and thermodynamics than COOH* formation over $O_V$-CeO$_2$, Cu$^+$-CeO$_{2-x}$ and Zn/Cu$^+$-CeO$_{2-x}$ catalysts, indicating that the doping of Cu and Zn into CeO$_2$ matrix could inhibit RWGS reaction via COOH* intermediate and hence enhances methanol selectivity via HCOO* intermediate. Meanwhile,

CO$_2$ activation over Cu$^+$-CeO$_{2-x}$ catalyst is more kinetically favorable than that over $O_V$-CeO$_2$ catalyst. Moreover, a small amount of Zn added into CuCe catalyst to form Zn/Cu$^+$-CeO$_{2-x}$ catalyst significantly enhances the activity of CO$_2$ activation to HCOO* and therefore promotes subsequent reactions to product CH$_3$OH. As shown in Fig. 6d, CO$_2$ binding energy has a good linear relationship with the activation barrier of CO$_2$ hydrogenation to HCOO* over $O_V$-CeO$_2$, Cu$^+$-CeO$_{2-x}$ and Zn/Cu$^+$-CeO$_{2-x}$ catalysts, in which Zn/Cu$^+$-CeO$_{2-x}$ catalyst exhibits high CO$_2$ binding energy, and greatly lowers the activation barrier of CO$_2$ hydrogenation to HCOO*. Thus, Zn/Cu$^+$-CeO$_{2-x}$ catalyst facilitates the formation of HCOO* intermediate in comparison with the $O_V$-CeO$_2$ and Cu$^+$-CeO$_{2-x}$ catalysts.

Starting from the HCOO* intermediate, the free energy diagram of CO$_2$ hydrogenation to methanol over Zn/Cu$^+$-CeO$_{2-x}$ catalyst is further investigated, as depicted in Fig. 7. HCOO* hydrogenation to H$_2$COO* is more favorable compared with HCOOH* formation in kinetics (63.1 *vs.* 131.2 kJ·mol$^{-1}$). Both the HCOOH* and H$_2$COO* intermediates can further be hydrogenated to H$_2$COOH* with the reaction energies of −22.4 and −1.4 kJ·mol$^{-1}$, respectively. Interestingly, our results show that H$_2$COO* hydrogenation to H$_2$COOH* is a spontaneous reaction without any barrier over Zn/Cu$^+$-CeO$_{2-x}$ catalyst. Subsequently, the decomposition of H$_2$COOH* to H$_2$CO* is exothermic by 18.5 kJ·mol$^{-1}$, however,

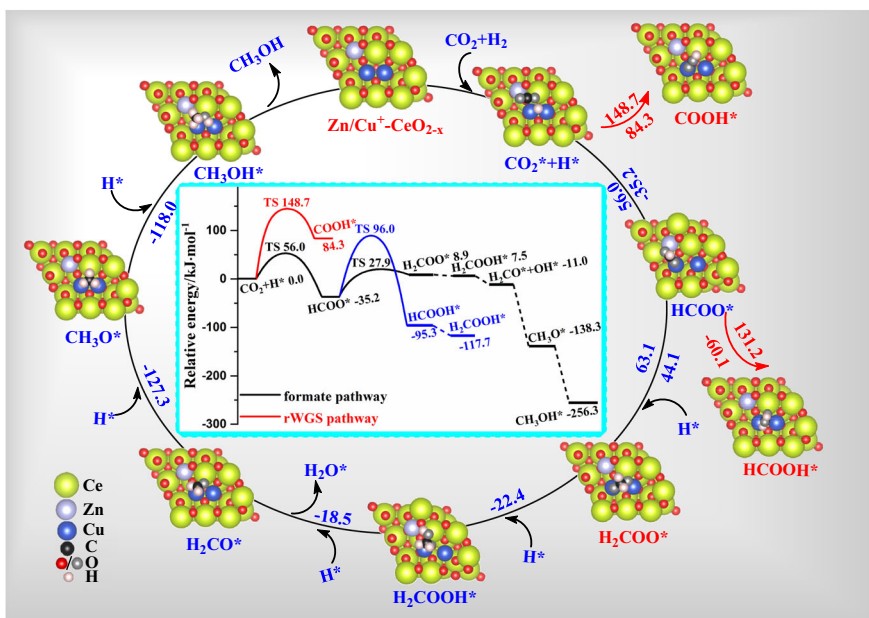

**Fig. 7 | The free energy diagram and the corresponding structure of the potential reaction pathway for CO₂ hydrogenation to methanol over Zn/Cu⁺-CeO₂₋ₓ catalyst.** The color codes, RWGS, and TS are the same as those in Fig. 6.

$H_2CO^*$ successive hydrogenation to $CH_3OH^*$ via $CH_3O^*$ intermediate is strongly exothermic by 127.3 and 118.0 kJ·mol⁻¹, respectively. Overall, the formation of $CH_3OH$ over $Zn/Cu^+$-$CeO_{2-x}$ catalyst mainly undergoes the pathway of $HCOO^* + H^* \rightarrow H_2COO^* + H^* \rightarrow H_2COOH^* \rightarrow H_2CO^* + OH^* \rightarrow CH_3O^* + H^* \rightarrow CH_3OH^*$, which is kinetically and thermodynamically preferred compared to other possible pathways. Thus, DFT calculations well explain the experimental activity results.

## Discussion

A MOFs crystal engineering strategy was employed to develop a series of CuZnCe catalysts for methanol synthesis. A ternary metal Ce-CuZn catalyst was shown to possess high STY (400.3 g·kg$_{cat}$⁻¹·h⁻¹) and selectivity of 71.1% to methanol (comparable to the commercial CuZnAl system). In particularly, the developed catalyst demonstrated long-term stability when tested for 170 h on stream. It was found that the order of introduction of metal during MOFs preparation influences the growth of MOFs and hence the distribution of active Cu phases in the final catalyst. The Ce-MOF was grown first and then in-situ ion exchange of $Cu^{2+}$ and $Zn^{2+}$ ions into Ce-MOF was used to derive Ce-CuZn-MOF. After its pyrolysis, the obtained Ce-CuZn catalyst with many active $Cu/Zn$-$O_V$-$Ce$ species thus presented robust property for $CO_2$ hydrogenation to methanol with excellent stability, which was comparable to that of the industrial CuZnAl catalyst.

In-situ DRIFTS experiment and DFT calculations provided insights into the reaction mechanism of methanol synthesis, proceeding mainly through the formate path. Methanol synthesis is a structure sensitive reaction and CO-DRIFTS was used to show that the surface composition of different Cu species influenced catalytic activity. Compared to $O_V$-$CeO_{2-x}$ catalyst, the $Cu^+$-$CeO_{2-x}$ catalyst with Cu doping into $CeO_{2-x}$ reduces the activation barrier of $CO_2$ hydrogenation to generate $HCOO^*$. Furthermore, doping Zn into $Cu^+/CeO_{2-x}$ can largely facilitate $H_2$ dissociation and the formation of $HCOO^*$. A similar Zn promotion effect on $CO_2$ hydrogenation to $CH_3OH$ has already been reported[21], and the addition of Zn in the Cu-Zn-Ce oxide catalysts was beneficial to inhibit the RWGS reaction. As a result, $CO_2$ hydrogenation to methanol is preferred to occur by the formate pathway, and $CH_3OH$ selectivity can be improved.

In summary, the Ce-CuZn catalyst outperformed by large amount the other catalysts is attributed to the following reasons: The Ce-CuZn catalyst from atomic-level substitution of Cu and Zn into Ce-MOF precursor produced many active $Cu/Zn$-$O_V$-$Ce$ species. Moreover, the Ce-CuZn catalyst has abundant $Cu^+$ species, a large number of oxygen vacancies, and more weaker basic sites for $CO_2$ adsorption. Kumari et al. reported that the number of oxygen vacancies could influence $CO_2$ activation via lower reaction barriers of $CO_2$ dissociation[55]. Our DFT calculation results have shown that the incorporation of Cu and Zn into $CeO_2$ with abundant oxygen vacancies can facilitate the $H_2$ dissociation and the formation of $HCOO^*$, thus improving $CO_2$ hydrogenation over Ce-CuZn catalyst via formate intermediates. In addition, it is worth noting that oxygen vacancies in $Cu/CeO_2$ are not beneficial for CO hydrogenation because they are poisoned by adsorbed $CO_2$ to form carbonate-like species[40]. Thus, the role of oxygen vacancies and their stability in $CO/CO_2$ hydrogenation still needs to be investigated in the development of new catalysts.

This work provides an atomic level regulating strategy towards constructing multi-metal catalysts step by step with effective active sites for $CO_2$ hydrogenation to methanol. The synergistic effect of CuZnCe in the $Cu/Zn$-$O_V$-$Ce$ species catalyses $CO_2$ activation and the hydrogenation of formates to achieve a high yield of methanol. It was illustrated that the active sites for methanol synthesis would be affected by the kind of metal species, preparation method, and the order of introduction of metal. This approach of precisely engineering active sites by controlling the structure and atom vacancies of the catalyst can be extended to other catalytic systems.

In the future, the metal components should be optimized to consider their economy as the Cu loading has exceeded 50%. The methanol yield can be further improved as the CO selectivity is still high over CuZnCe catalysts. Thus, more strategies should be adopted to suppress the reverse water-gas shift reaction. In addition, as the polymetallic $Cu/Zn$-$O_V$-$Ce$ active sites have presented excellent catalytic performance, the high-entropy alloy-based catalysts with various unique synergistic effects may have potential application in methanol synthesis, but it would be challenging to illustrate the polymetallic interfaces.

## Methods
### Chemicals
Cerium nitrate hexahydrate (Ce(NO₃)₃·6H₂O, AR), copper nitrate trihydrate (Cu(NO₃)₂·3H₂O, AR), zinc nitrate hexahydrate

(Zn(NO₃)₂·6H₂O, AR), methanol (CH₃OH, AR), and ethanol (C₂H₅OH, AR) were purchased from Sinopharm Chemical Reagent Co., Ltd. 1,3,5-benzenetricarboxylic acid (1,3,5-BTC, 98%) was purchased from Aladdin. Deionized water (DI) was prepared in the laboratory. All chemicals were used without further treatment.

## Catalyst preparation
The preparation of CuZnCe catalysts is described in Supplementary Fig. 1 and depicted below.

Synthesis of Ce-CuZn (Route 3): The Ce-MOF was synthesized first, and then Ce-CuZn·MOF was prepared. Specifically, 1.736 g of Ce(NO₃)₃·6H₂O and 0.840 g of 1,3,5-BTC were dissolved into a solvent mixture consisting of 40 mL methanol and 40 mL ethanol, respectively. Then the 1,3,5-BTC solution was added into the cerium nitrate solution, followed by the addition of 40 mL H₂O. After stirring at RT for 20 min, the mixture was centrifuged. The supernatant was discarded and the solid obtained from centrifugation was washed with ethanol several times. The product was dried at 60 °C in a vacuum oven for 9 h and in a common oven for another 3 h to obtain the dried Ce-MOF. 0.8 g of the above dried Ce-MOF powder was dispersed into 12 mL of methanol, then a solution of 0.484 g of Cu(NO₃)₂·3H₂O and 0.298 g of Zn(NO₃)₂·6H₂O dissolved in 15 mL methanol was added into the above Ce-MOF suspension under ultrasonic conditions. After sonication for 30 seconds, the mixture was allowed to stand for 20 min and then centrifuged and washed with ethanol several times. Finally, the mixture was dried at 60 °C for 12 h in a vacuum oven for 9 h and in a common oven for another 3 h and then calcined at 450 °C for 5 h in air with a heating rate of 2 °C/min. The obtained samples were denoted as Ce-CuZn·MOF and Ce-CuZn, respectively. In addition, the synthesis of Ce-Cu·MOF and Ce-Zn·MOF was similar to that of the Ce-CuZn·MOF except that the zinc nitrate or copper nitrate was not added, respectively, and the final catalysts were named Ce-Cu and Ce-Zn catalysts.

The preparation of other CuZnCe series catalysts is presented in the Supplementary Information.

## Catalyst characterization
All the detailed catalyst characterizations are depicted in the Supplementary Information.

## DFT calculations
The details of the DFT calculations are shown in the Supplementary Information.

## Catalyst evaluation
The CuZnCe catalysts were evaluated in a fixed-bed reactor under reaction conditions of 2.0 - 2.8 MPa, 10,000 - 20,000 mL·g$_{cat}^{-1}$·h⁻¹, 220 - 320 °C, and a H₂: CO₂: N₂ ratio of 72: 24: 1. The detailed catalyst evaluation method is described in the Supplementary Information.

## Data availability
The data supporting the findings of this study are available within the main text, the Supplementary Information file, the Source Data files, or from the corresponding authors upon request. Source data are provided with this paper and also deposited in Figshare repository (https://doi.org/10.6084/m9.figshare.25129247). Source data are provided with this paper.

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

## Acknowledgements

This work was supported financially by the Natural Science Foundation of Jiangxi Province for Distinguished Young Scholars (20232ACB213001, R.Y.), National Natural Science Foundation of China (U23A20132, R.Z.; 22005296, R.Y.), National Key R&D Program of China (2022YFA1504503, J.L.), Science Foundation for Distinguished Young Scholar of Shanxi Province (20210302121005, R.Z.), Natural Science Foundation of Chongqing, China (CSTB2022NSCQ-MSX0231, R.Y.), Natural Science Foundation of Jiangxi Province (20224BAB213015, R.Y.), and the Thousand Talents Plan of Jiangxi Province (jxsq2023101072, R.Y.). The authors would also like to thank Dr. Caixia Meng from Dalian Institute of Chemical Physics, Chinese Academy of Sciences for the quasi in-situ XPS analysis and Shiyanjia Lab (www.shiyanjia.com) for the TEM test. A.G. thanks the support from the U.S. Department of Energy, Office of Science, Office of Basic Energy Sciences, Chemical Sciences, Geosciences, and Biosciences Division, Catalysis Science Program to the SUNCAT Center for Interface Science and Catalysis.

## Author contributions

R.Z., M.D., Z.J. and J.L. supervised the project. R.Y. and M.D. conceived and performed the experiments. L.M., B.W. and R.Z. performed DFT calculations and made their interpretations. X.W. carried out the SEM measurement. J.M. and Z.J. performed the EXAFS analysis. X.H. and Y.M. performed in-situ DRIFTS characterizations. A.G. and W.L. helped to interpret the EXAFS, TEM and XPS data. R.Y., L.M. and J.M. wrote the draft and the other authors revised the manuscript.

## Competing interests

The authors declare no competing interests.
