## [Peer Review File · Nature Communications]

A Ce-CuZn catalyst with abundant Cu/Zn-OV-Ce active sites for CO₂ hydrogenation to methanolREVIEWER COMMENTS

Reviewer #1 (Remarks to the Author):

Manuscript ID:

Title: High-performance Ce-CuZn catalyst with abundant Cu/Zn-Ov-Ce active sites for CO₂ hydrogenation to methanol

Reviewer's comments: resubmit after major revision

The study synthesizes a set of Ce-CuZn catalyst with abundant Cu/Zn-Ov-Ce active sites, and further utilizes them as photocatalysts for CO₂ hydrogenation to methanol. It focuses on an important environmental topic within the journal scope. But there are still many problems to be solved. It is my opinion that this work will need to be resubmitted after major revisions. Some concerns are as follows:

1. In the section of synthetic route, the authors indicated that the calcination temperature of the sample was 450 °C. Can the structural stability of Ce-MOF be maintained under these conditions?
2. From Figure 1(c), only lattice streaks of CeO₂ were observed, and the surface scanning results showed that a solid solution was formed. However, can the solid solution be stabilized after reduction by H₂, and was Cu accompanied by migratory agglomeration during this process? Please give relevant evidence.
3. Since oxygen vacancies were the main active species in this work, but the Raman results in Figure 2(c) showed a more pronounced difference in oxygen vacancy signals, but it didn't seem to match well with the oxygen vacancy results at 531 eV in XPS spectra? Also, the Ce³⁺/(Ce³⁺+Ce⁴⁺) ratio results in table S1 did not differ much, was it a computational error and the oxygen vacancies were not actually present in large numbers? Are there other means of proving the true existence of oxygen vacancies?
4. The order in which the elements Cu, Zn, and Ce are paired, and the order in which the ions are introduced are not all verified. The reasonableness of the claim of Ce-CuZn is to be examined.
5. The DFT results were meaningful only under the condition that the model was built to match the real situation of the catalyst. This work must provide the basis for building the doping model, which can be explained by experiment of calculation.
6. The performance data is really good from a pure perspective. However, one point to consider is that the content of Cu has exceeded 50%. So the economy and applicability of this system is to be examined. Secondly, the reaction conditions of this experiment are not exactly the same as those of other experiments, and it is meaningless to claim that it is STY maximum.
7. Further refinement of the manuscript graphics and tables are needed.

Reviewer #2 (Remarks to the Author):

This work reports a high-performance Ce-CuZn catalyst with abundant Cu/Zn-OV-Ce active sites for CO₂ hydrogenation to methanol. In recent years, the CO₂ hydrogenation as an effective method of CCUS is a very hot research topic that attract many readers. A robust Ce-CuZn catalyst in this study was achieved by a MOFs engineering method and the deep structure-performance relationship has been illustrated well. This is a systematic and innovative study work combined well with the DFT calculation. Many

control experiments are carefully done and the conclusions are fully supported by the data presented. This work is meaningful to construct efficient multi-metal catalysts. Therefore, I would like to recommend publication of this manuscript in Nature Communications after addressing the following questions:

1. It seems that the oxygen vacancy in the Cu/Zn-OV-Ce active sites is a significant factor for methanol synthesis. How did the oxygen vacancies produce over the Ce-CuZn catalysts? The formation reason should be provided. The Raman spectra was used to study the oxygen vacancies in this work. In my opinion, more characterization experiments like the EPR should be done to confirm the oxygen vacancies.
2. In Table 1, it shows that the Cu loading is more than 50%, however, why the Cu lattice fringes was not observed in Figure 1c?
3. In Table 1, why did the control catalyst Ce-CuZn-IM exhibit much lower Cu loading and higher Zn loading than the Ce-CuZn catalyst? It suggests to control similar metal loading with Ce-CuZn catalyst and then to compare their catalytic performance. This control experiment should be added.
4. A mistake is shown in the caption of Figure 2. There are two (c) captions in Figure 2. The H₂-TPR should be deleted as it has been shown in Figure S8.
5. In Figure 2a, the diffraction peaks belong to CuO species is very obvious, however, why the CO adsorbed on CuO species were not be observed in Figure 5a?
6. In Figure 4a, the CuZnCe, CuZn-Ce and Ce-CuZn samples with similar metals but presented distinct catalytic performance. The authors should provide more explanation to illustrate their difference.
7. Authors have used in-situ CO-DRIFTS to investigate the surface interactions with the CO probe molecule, and further confirmed the existence of Cu⁺ species over CuZnCe catalysts. Whether it can be verified the CO adsorption behaviors from DFT calculations?
8. The legend in Figure 6a seems wrong, only the differential charge density ($\Delta\rho$) of Cu⁺-CeO_{2-x} and Zn/Cu⁺-CeO_{2-x} catalysts are given in, while the Cu²⁺-CeO_{2-x} catalyst was missing.
9. Overall the manuscript is well written, could the authors provide more perspective on this research in the conclusion section?

Reviewer #3 (Remarks to the Author):

This manuscript, by Ye et al., studies the synthesis of CuZnCe catalysts using a MOF precursor. One of the catalysts shows a remarkable activity for methanol formation via CO₂ hydrogenation, comparable to technical CuZnAl catalysts.

My general opinion is that in spite of all the characterizations performed, it is still not clear why the Ce-CuZn catalyst outperforms by large amount the other catalysts. The Zn-Ov-Ce sites that would explain its higher activity could also be present on the Ce-CuZn-IM. I think that more catalysts have to be prepared to assert which is the cause of the higher activity observed. The Zn content is too different (1% vs. 11%) and could have an effect. In fact, one important work that the authors do not discuss (Hensen et al, ACS Catal. 2021, 11, 4880–4892) shows that Zn content has a dramatic effect on CuZnCe catalysts for this reaction. I recommend to compare the results with that paper to enhance the discussion.

Authors claim that the peak at 183°C observed in the TPR curves indicates the interaction of Cu-Zn and Cu-Ce, which is related to oxygen vacancies. If this is correct, the Ce-CuZn-IM would also have an important amount of Zn-Ov-Ce sites. Since it contains more Zn than Ce-CuZn, Ce-CuZn-IM would also have more vacancies related to Zn. How this correlates with the poor activity of Ce-CuZn-IM catalyst? XAS analysis. The main finding is that, for Ce-CuZn, "...we speculate that some Cu and Zn were incorporated into CeO₂ lattice to form Cu/Zn-OV-Ce species while the other Cu and Zn on the catalyst surface combined with CeO₂ to form Cu/Zn-ceria interfaces..." This result could be valid also for Ce-CuZn-IM? For this catalyst, the fact that there is more Zn (11% vs. 1% for Ce-CuZn) could facilitate the formation of CuZn alloy but also a part of Ce could be forming the Cu/Zn-OV-Ce species.

CO adsorption monitored by IR. In line 390: "...In addition, the Ce-CuZn and CuZn-Ce samples present an obvious shoulder peak at 2138 cm⁻¹, which belongs to single-site Cu⁺/Zn-O-Ce species..." How this is possible if CuZn-Ce has a very tiny amount of Zn? Please discuss it. It is possible that this peak is related only to Cu⁺-Ov-Ce³⁺ vacancies?

How is the selectivity to CO of Ce-CuZn compared to technical CuZnAl?

Why is the data for CuZn missing from table 1?

In situ DRIFTS analysis at steady state show the typical surface species that previous authors have shown in these type of catalysts. The presence of these surface species is not a direct evidence of a determined pathway for methanol formation. More experiments are required (transient for example) to assess the role of each surface species in the mechanism and discard spectators.

In the introduction is discussed that some authors found that Cu-Ov-Ce active species could improve the CO₂ transformation to methanol. The Graciani et al. reference did not show that because it deals with model systems composed of CeO_x/Cu(111). Moreover, some authors have proposed that vacancies in Cu/CeO₂ are not beneficial because they are poisoned by adsorbed CO₂ (ACS Catal. 2020, 10, 11532–11544)

Responses to Reviewers' Comments

Reviewer #1 (Remarks to the Author):

Manuscript ID:

Title: High-performance Ce-CuZn catalyst with abundant Cu/Zn-Ov-Ce active sites for CO₂ hydrogenation to methanol

*Reviewer's comments: **resubmit after major revision***

The study synthesizes a set of Ce-CuZn catalyst with abundant Cu/Zn-Ov-Ce active sites, and further utilizes them as photocatalysts for CO₂ hydrogenation to methanol. It focuses on an important environmental topic within the journal scope. But there are still many problems to be solved. It is my opinion that this work will need to be resubmitted after major revisions. Some concerns are as follows:

Response: We would like to thank the reviewer for his/her valuable comments.

1. In the section of synthetic route, the authors indicated that the calcination temperature of the sample was 450 °C. Can the structural stability of Ce-MOF be maintained under these conditions?

Response: Thank you for this comment on the structural stability of Ce-MOF, which prompted us to extend our discussion in our revised manuscript and SI. As shown in the **Figure S1c**, the Ce-MOF and Ce-CuZn-MOF are decomposed under the calcination conditions. The dried Ce-MOF sample lost the solvent molecules at about 150 °C and then lost the 1,3,5-BTC ligand at about 350 °C. The decomposition temperatures for Ce-CuZn-MOF are lower than Ce-MOF because Ce-CuZn-MOF was prepared via more steps including solvent washing. The CuZnCe catalysts reported here are derived from MOFs used as precursors. The MOFs with tunable chemical components and tailored structures are an ideal platform to engineer the active sites from the atomic and molecular levels. Thus, the Ce-CuZn catalyst with enriched Cu/Zn-Ov-Ce active sites has been fabricated through the atomic-level substitution of Cu and Zn into Ce-MOF precursor and then decomposed to Ce-CuZn catalyst.

Figure S1c. TG curves of dried Ce-MOF and Ce-CuZn-MOF samples under the air atmosphere.

We have now conducted thermogravimetric analysis (TGA) and added the following contents in the Results and Supplementary Information:

Page 7: “After calcination, the diffraction peaks of CuO and CeO₂ could be found, showing the segregation and agglomeration of metal species as well as the decomposition of MOF precursors. The dried Ce-MOF sample lost the solvent molecules at about 150 °C and then lost the 1,3,5-BTC ligand at about 350 °C (Figure S1c). The decomposition temperatures for Ce-CuZn-MOF are lower than Ce-MOF because that Ce-CuZn-MOF was prepared via more steps including solvent washing.”

Page S6: “*TG test:* About 10 mg of sample was filled into a crucible and then it was heated to 800 °C with a heating rate of 5°C/min under the air atmosphere. The set-up type for thermogravimetric (TG) test is Mettler Toledo TGA/DSC 3+.”

2. From Figure 1(c), only lattice streaks of CeO₂ were observed, and the surface scanning results showed that a solid solution was formed. However, can the solid solution be stabilized after reduction by H₂, and was Cu accompanied by migratory agglomeration during this process? Please give relevant evidence.

Response: Firstly, Figures 1c-1g are post-reduction (by H₂) HRTEM images of the Ce-CuZn sample, thus indicating that the solid solution can be stabilized after reduction. We have now performed Raman and EPR to further probe the solid solution structure of the reduced Ce-CuZn sample. The

Raman and EPR tests were also performed on the reduced samples. An obvious broad band (550~650 cm^{-1}) induced from oxygen vacancies was observed over CuZn-Ce and Ce-CuZn samples (**Figure 2c**), which was attributed to the substitutional incorporation of Cu/Zn ions into the CeO_2 lattice. The EPR spectrum of Ce-CuZn sample presents obvious oxygen vacancies peak at g_{\parallel} value of 1.99, which is much different with the other two samples (**Figure 2f**). Moreover, the type K peaks at g_{\parallel} values of 2.29 and 1.83 are ascribed to $\text{Cu}^{2+}/\text{Zn}^{2+}$ dimer, which could be observed when two neighboring Ce^{4+} ions with short separation distance are substituted by $\text{Cu}^{2+}/\text{Zn}^{2+}$ ions³. Thus, the appearance of K signals suggests that Cu/Zn-Ov-Ce solid solution is indeed kept in the reduced Ce-CuZn sample.

Figure 2c. Raman spectra of reduced CuZnCe samples.

Figure 2f. EPR spectra of the CuZnCe catalysts.

Regarding the reviewer's point "was Cu accompanied by migratory agglomeration during this process?", we have now supplemented the TEM results on the calcined Ce-CuZn sample, as shown in **Figures S4-S5**. The mean Cu nanoparticle sizes of calcined (5.34 nm) and reduced (8.17 nm) Ce-CuZn samples indicate that Cu nanoparticles were accompanied by slight migratory agglomeration during the reduction process.

Figure S4. TEM (a, b) and HRTEM (c) images of calcined Ce-CuZn sample with corresponding elemental mapping.

Figure S5. TEM images of (a) calcined and (b) reduced Ce-CuZn samples with corresponding particle size

distributions.

The added contents in the Results and Supplementary Information are as follows:

Page 9: "It should be mentioned that the mean Cu nanoparticle sizes were increased from 5.34 nm over the calcined Ce-CuZn to 8.17 nm over the reduced Ce-CuZn, indicating that the Cu nanoparticles were accompanied by slight migratory agglomeration during reduction process (Figures S4-S5). However, the solid solution could be kept over the reduced Ce-CuZn sample, which would be demonstrated by the following Raman and electron paramagnetic resonance (EPR) results."

3. *Since oxygen vacancies were the main active species in this work, but the Raman results in Figure 2(c) showed a more pronounced difference in oxygen vacancy signals, but it didn't seem to match well with the oxygen vacancy results at 531 eV in XPS spectra? Also, the $Ce^{3+}/(Ce^{3+}+Ce^{4+})$ ratio results in table S1 did not differ much, was it a computational error and the oxygen vacancies were not actually present in large numbers? Are there other means of proving the true existence of oxygen vacancies?*

Response: Thank you for this comment on the oxygen vacancies. The previous XPS test was done under *ex-situ* conditions. To confirm the data, we have supplemented the *quasi in-situ* XPS of reduced catalysts and the results are shown in **Figures S11-S12** and **Table S1**. O1s peaks are deconvoluted into three components (**Figure S12b**), namely lattice oxygen (O_{α}), oxygen vacancies (O_{β}) and surface oxygen (O_{γ})^{1,2}. Similarly, the Ce 3d peaks are deconvoluted into four peaks of $3d^{10}4f^1$ Ce^{3+} (u_0 , u_1 , u'_0 , and u'_1) and six peaks of $3d^{10}4f^0$ Ce^{4+} (v_0 , v_1 , v_2 , v'_0 , v'_1 , and v'_2) (**Figure S12c**). The Ce-CuZn catalyst exhibits higher ratio (29.9%) of O_{β} (the peak of oxygen vacancies) than the other two samples (**Table S1** and **Figure S12b**), indicating that Ce-CuZn catalyst possesses higher concentration of oxygen vacancies. However, the $Ce^{3+}/(Ce^{3+}+Ce^{4+})$ ratio (34.8%) of Ce-CuZn is slightly lower than that of CuZn-Ce (36.4%), but higher than that over Ce-CuZn-IM (27.4%, **Table S1**). This is because the catalyst has two types of oxygen vacancies: I) generation from the reduction of Ce^{4+} to Ce^{3+} , and II) the replacement of Ce^{4+} by Cu/Zn ions, resulting in the formation of oxygen vacancies³. Thus, the Ce-CuZn catalyst with slightly lower $Ce^{3+}/(Ce^{3+}+Ce^{4+})$ ratio could still exhibit higher $O_{\beta}/(O_{\alpha}+O_{\beta}+O_{\gamma})$ ratio due to the substitutional incorporation of Cu/Zn ions into the CeO_2 lattice. Therefore, the oxygen vacancies results at 531.2 eV in O 1s spectra and the $Ce^{3+}/(Ce^{3+}+Ce^{4+})$ ratio in Ce 3d spectra are consistent with the Raman results.

Figure S11. *Quasi in-situ* XPS profiles of the reduced CuZnCe catalysts: (a) Cu2p, (b) Zn2p.

Figure S12. *Quasi in-situ* XPS profiles of the reduced CuZnCe catalysts. (a) Cu LMM, (b) O1s, (c) Ce3d.

Table S1. Surface Cu and Zn species of the reduced CuZnCe catalysts determined by *quasi in-situ* XPS.

Samples	$\text{Cu}^+ / (\text{Cu}^0 + \text{Cu}^+)$ (%)	$\text{O}_\beta / (\text{O}_\alpha + \text{O}_\beta + \text{O}_\gamma)$ (%)	Ce^{3+} BE (eV)	Ce^{4+} BE (eV)	$\text{Ce}^{3+} / (\text{Ce}^{3+} + \text{Ce}^{4+})$ (%)
CuZn-Ce	52.1	25.8	881.4, 884.8 898.5, 903.2	882.5, 888.8, 897.7 900.9, 907.4, 916.7	36.4
Ce-CuZn	56.5	29.9	881.4, 884.6 898.4, 902.9	882.3, 888.6, 897.7 900.8, 907.3, 916.6	34.8
Ce-CuZn-IM	—	21.3	881.4, 884.8 898.8, 902.9	882.6, 889.0, 898.1 901.0, 907.6, 916.8	27.4

To respond to “Are there other means of proving the true existence of oxygen vacancies?”, we have done another two tests including electron paramagnetic resonance (EPR) and chemisorption measurements, which display many obvious differences among the three catalysts. The EPR results and analysis are illustrated in the response to the first question of **Reviewer #2**. The chemisorption measurement to determine the amount of oxygen vacancies (N_{OV}) was developed by Zhu *et al.*⁴, which is depicted below.

Page S7 of Supplementary Information: “The oxygen vacancies tests were also performed on Autochem II 2920 instrument by one CO₂ adsorption, two N₂O pulsing titrations and two H₂-TPR processes in one sample test. The CO₂ adsorption was used to block oxygen vacancies in CeO₂-based metal catalysts. This method was referenced from Zhu *et al.*, who reported to determine the amount of oxygen vacancies (N_{OV}) over Cu/CeO₂ catalysts². Namely, the sample (~100 mg) was firstly pretreated at 350 °C for 1 h under the 10%H₂-Ar flow followed by cooling to 50 °C in a Ar flow. Next, a 5% N₂O-He gas mixture was periodically injected to the sample using a 0.58 mL sample loop until no N₂O consumption was observed. This N₂O consumption amount is denoted as $n_{N_2O, 1}$. Before the second N₂O experiment, the same sample was purged by Ar for 0.5 h and then reduced at 350 °C for 1 h under the 10%H₂-Ar flow followed by cooling to 50 °C in a Ar flow again. After cooling to 50 °C in a Ar flow, pulses of pure CO₂ were injected to the sample to block the ceria oxygen vacancies. After the CO₂ pulsing, the same N₂O pulsing experiment was carried out to titrate the remaining metallic copper sites. This N₂O consumption amount is denoted as $n_{N_2O, 2}$. The N₂O consumption by oxygen vacancies was derived from the difference between these two N₂O titration experiments and the ratio of N₂O to N_{OV} was set as 1. Finally, the amount of oxygen vacancies (N_{OV}) was calculated by $N_{OV} = \frac{n_{N_2O, 1} - n_{N_2O, 2}}{m_{cat}}$ ”

The above oxygen vacancies tests results are summarized in **Table 1**. It presents that the Ce-CuZn sample exhibits 18.1 $\mu\text{mol/g}_{\text{cat}}$ of oxygen vacancies, which is still much higher than the other two samples (2.9~6.0 $\mu\text{mol/g}_{\text{cat}}$). These results are consistent with the *quasi in-situ* XPS (**Table S1**) and the EPR results (**Figure 2f**).

Table 1. The physicochemical properties of CuZnCe catalysts.

Catalysts	Cu ^{a)} (wt.%)	Zn ^{a)} (wt.%)	Ce ^{a)} (wt.%)	S _{BET} (m ² /g)	Pore size (nm)	Pore volume (cm ³ /g)	S _{Cu} ^{b)} (m ² /g)	D _{Cu} ^{b)} (%)	N _{OV} ($\mu\text{mol/g}_{\text{cat}}$) ^{b)}
CuZn	80.65	3.55	-	2.9	26.9	0.01	3.7	0.7	-
CuZn-Ce	52.33	0.03	28.71	41.5	6.7	0.07	31.1	9.2	6.0
Ce-CuZn	52.82	1.12	31.74	40.2	12.1	0.12	23.1	6.7	18.1
Ce-CuZn-IM	22.36	11.47	48.80	41.4	7.2	0.08	22.0	15.2	2.9

^{a)} Metal loading results from ICP.

b) Metallic copper surface area (S_{Cu}), copper dispersion (D_{Cu}) and oxygen vacancies (N_{OV}) determined by N_2O titration and H_2 temperature-programmed reduction (H_2 -TPR).

Figure 2f. EPR spectra of the CuZnCe catalysts.

References

1. Liu, S. *et al.* Producing ultrastable Ni-ZrO₂ nanoshell catalysts for dry reforming of methane by flame synthesis and Ni exsolution. *Chem Catal.* **2**, 2262-2274 (2022).
2. Cheng, X. *et al.* NO reduction by CO over copper catalyst supported on mixed CeO₂ and Fe₂O₃: Catalyst design and activity test. *Appl. Catal. B: Environ.* **239**, 485-501 (2018).
3. Jiang, J., Yang, H., Jiang, H., Hu, Y. & Li, C. Boosting catalytic activity of Cu-Ce solid solution catalysts by flame spray pyrolysis with high Cu⁺ concentration and oxygen vacancies. *Chem. Eng. J.* **471**, 144439 (2023).
4. Zhu, J. *et al.* Mechanism and nature of active sites for methanol synthesis from CO/CO₂ on Cu/CeO₂. *ACS Catal.* **10**, 11532-11544 (2020).

The added contents in the Results and Supplementary Information are as follows:

Page 14: “*Quasi in-situ* X-ray photoelectron spectroscopy (XPS) was further performed to analyze the surface species over the reduced CuZnCe series samples. The binding energy of Cu 2p_{3/2} over the Ce-CuZn-IM present at 932.8 eV is attributed to Cu⁰/Cu⁺ species (Figure S11a). Moreover, the Cu LMM XAES spectra were carried out to determine the specific Cu⁺/(Cu⁺+Cu⁰) ratio as shown in Figure S12a and Table S1. The Ce-CuZn sample has a higher Cu⁺ content (56.5%) and that over Ce-CuZn-IM is not available due to the effect of Zn 2p.” “Moreover, O1s peaks are deconvoluted into three components (Figure S12b), namely lattice oxygen (O_α), oxygen vacancies (O_β) and surface oxygen (O_γ). Similarly, the Ce 3d peaks are deconvoluted into four peaks of 3d¹⁰4f¹ Ce³⁺ (u₀, u₁, u₀, and u₁) and six peaks of 3d¹⁰4f⁰ Ce⁴⁺ (v₀, v₁, v₂, v₀, v₁, and v₂) (Figure S12c). The Ce-CuZn catalyst exhibits a higher ratio (29.9%) of O_β than the other two samples (Table S1), indicating that Ce-CuZn catalyst possesses a

higher concentration of oxygen vacancies. However, the $\text{Ce}^{3+}/(\text{Ce}^{3+}+\text{Ce}^{4+})$ ratio (34.8%) of Ce-CuZn is slightly lower than that over CuZn-Ce (36.4%), but higher than that over Ce-CuZn-IM (27.4%, Table S1). This is because that the catalyst has two types of oxygen vacancies: I) generation from the reduction of Ce^{4+} to Ce^{3+} , and II) the replacement of Ce^{4+} by Cu/Zn ions, resulting in the formation of oxygen vacancies. Thus, the Ce-CuZn catalyst with slightly lower $\text{Ce}^{3+}/(\text{Ce}^{3+}+\text{Ce}^{4+})$ ratio could still exhibit higher $\text{O}_\beta/(\text{O}_\alpha+\text{O}_\beta+\text{O}_\gamma)$ ratio due to the substitutional incorporation of Cu/Zn ions into the CeO_2 lattice. Therefore, the oxygen vacancies results at 531.2 eV in O 1s spectra and the $\text{Ce}^{3+}/(\text{Ce}^{3+}+\text{Ce}^{4+})$ ratio in Ce 3d spectra are consistent with the Raman results.”

Page 15: “Moreover, EPR and chemisorption measurements were carried out to confirm the oxygen vacancies. The EPR spectrum of Ce-CuZn sample presents obvious oxygen vacancies peak at g_{\parallel} value of 1.99, which is much different with the other two samples (Figure 2f). Moreover, the type K peaks at g_{\parallel} values of 2.29 and 1.83 are ascribed to $\text{Cu}^{2+}/\text{Zn}^{2+}$ dimer, which could be observed when two neighboring Ce^{4+} ions with short separation distance are substituted by $\text{Cu}^{2+}/\text{Zn}^{2+}$ ions. Thus, the appearance of K signals suggests that Cu/Zn- O_V -Ce solid solution is indeed generated in Ce-CuZn sample. In addition, the chemisorption measurement to determine the amount of oxygen vacancies (N_{OV}) was developed by Zhu *et al.*, and the results are summarized in Table 1. It presents that the Ce-CuZn sample exhibits 18.1 $\mu\text{mol}/\text{g}_{\text{cat}}$ of oxygen vacancies, which is still much higher than the other two samples (2.9~6.0 $\mu\text{mol}/\text{g}_{\text{cat}}$). These results are consistent with the above *quasi in-situ* XPS and the EPR results.”

Page S7: “The oxygen vacancies tests: ...”, which is also shown in the above response.

Page S9: “*Quasi in-situ* XPS test: Quasi in-situ X-ray photoelectron spectroscopy (XPS) experiments were carried out on Thermo Scientific Escalab 250Xi+. Before the test, the samples were pre-reduced in an atmospheric processing chamber at 350 °C for 1 h under the 20% H_2 -Ar atmosphere, which was interconnected with the ultra-high vacuum chamber of XPS analysis chamber to avoid the influence of air exposure. The set-up was equipped with a monochromatic Al K α X-ray radiation source ($h\nu = 1486.6$ eV). The beam diameter was 500 μm , and the acceleration voltage was 15 kV.”

Page S25:

Figure S11. *Quasi in-situ* XPS profiles of the reduced CuZnCe catalysts: (a) Cu2p, (b) Zn2p.

4. The order in which the elements Cu, Zn, and Ce are paired, and the order in which the ions are introduced are not all verified. The reasonableness of the claim of Ce-CuZn is to be examined.

Response: To answer this question, we have prepared six control samples and supplemented our data with another seven control samples. The pure CeO₂ support and the binary system of CuZn, Ce-Cu, and Ce-Zn catalysts have presented poor catalytic performance (**Table S4**). Thus, it is needed to prepare the ternary system. For the order of introducing the Cu, Zn, and Ce elements, they were firstly introduced together, but the CuZnCe sample exhibited only 1.4% of Con._{CO₂} at 280 °C. The CuZnCe sample only grew Ce-MOF with very low CuZn and shows a poor performance. Thus, the three metals could not be added together.

Which element should be introduced first? The Cu was introduced first, and the CuZn-Ce sample with extremely low content of Zn (0.03%, **Table 1**) presented low performance as the Zn was lost during the second step of preparation. When the Zn was introduced first, the Zn-CuCe sample was could not be synthesised because the Zn-MOF could not be generated under similar conditions. When the Ce was introduced first, the CeCu-Zn with low Cu content and CeZn-Cu samples with low Zn content resulted in poor performance because Ce-MOF could not be grown together with Cu/Zn-MOF. However, the Ce-CuZn sample could grow Ce-MOF and Cu/Zn-MOF well with two main steps, and thus it had high Cu/Zn-O_v-Ce species and exhibited the best performance. When the Zn content was increased, the Ce₂-CuZn₂ and Ce₁-CuZn₄ samples showed a decrease CO₂ conversion although the methanol yield was slightly increased.

In addition, we also investigated more complicated preparation procedures with three main steps. The Ce-Cu-Zn and Ce-Zn-Cu samples also have good performance, but the preparation procedures are more complicated. Furthermore, although the Sel._{MeOH} and STY_{MeOH} slightly increased over Ce₂-CuZn₂, Ce₁-CuZn₄ and Ce-Cu-Zn samples, they were increased at the expense of CO₂ conversion compared with Ce-CuZn sample. When the Con._{CO₂} over Ce-CuZn sample was also near 10%, the Sel._{MeOH} and STY_{MeOH} were increased to 45.5% and 154.0 g·kg_{cat}⁻¹·h⁻¹, respectively.

In summary, the optimized Ce-CuZn catalyst has the appropriate metal elements and suitable introduction order to form abundant Cu/Zn-O_v-Ce species, thus it presents the best catalytic performance for methanol synthesis compared with the control catalysts.

Table S4. Catalytic performance of the CuZnCe catalysts. ^{a)}

Catalysts	Con. CO ₂ %	Sel. CH ₃ OH %	STY _{CH₃OH} g·kg _{cat} ⁻¹ ·h ⁻¹	Remark
Ce	0.38	0	0	The CeO ₂ support without Cu or Zn has no activity.
CuZn	0.9	38.4	11.7	CuZn without CeO ₂ support would be sintering and has poor activity.
Ce-Cu ^{b)}	9.7	16.9	58.1	The sample without Zn promoter has low performance.
Ce-Zn ^{c)}	0.17	88.2	5.5	The sample without main Cu metal has bad performance.
CuZnCe	1.4	15.3	7.4	The sample only grew Ce with very low CuZn and has bad performance. The three metals could not be added together.
CuZn-Ce	4.6	22.1	36.0	Zn was lost during preparation and has low performance.
Ce-CuZn	15.1	26.4	140.6	The sample has high Cu/Zn-Ov-Ce species and exhibits the best performance.
CeCu-Zn	1.05	34.2	12.7	The sample with low Cu content has bad performance due to CeCu-MOF could not be grown together.
CeZn-Cu	5.3	20.5	38.6	The sample with low Zn content has bad performance due to CeZn-MOF could not be grown together.
Zn-CuCe	NA	NA	NA	The sample could not be prepared due to Zn-MOF could not be prepared under the similar conditions.
Ce ₂ -CuZn ₂	10.9	38.4	148.6	The samples with increased Zn content would decrease CO ₂ conversion although the methanol yield was slightly increased.
Ce ₁ -CuZn ₄	8.9	47.4	149.6	
Ce-Cu-Zn ^{d)}	10.7	43.3	165.7	The two samples also have good performance, but the preparation procedures are more complicated.
Ce-Zn-Cu ^{d)}	12.4	23.9	106.2	
Ce-CuZn-IM	1.6	46.1	26.8	The sample prepared with impregnation method (IM) has lower performance than Ce-CuZn.
Ce-CuZn-IM-B	3.0	42.6	56.7	The sample with similar Cu and Zn loading but still has lower performance than Ce-CuZn.

^{a)} Reaction conditions: 280 °C, 2 MPa, 10,000 h⁻¹, H₂/CO₂/N₂=72/24/1.

^{b)} “-” means two main steps for the sample preparation. The Ce-MOF was grown first and then the Cu-MOF was grown on the Ce-MOF. It is the same for other samples.

^{c)} Reaction temperature for this sample was 320 °C due to its extremely low performance.

^{d)} “- -” means three main steps for the sample preparation.

The added contents in the Results and Discussion are as follows:

Page 18: “To assess the effect on performance of the order in which the ions are introduced, more control samples

have been prepared and evaluated, as illustrated in Table S4. The pure CeO₂ support and the binary system of CuZn, Ce-Cu, and Ce-Zn catalysts have presented poor catalytic performance. Thus, it is necessary to prepare the ternary system. For the order of introducing the Cu, Zn, and Ce elements, they were firstly introduced together, but the CuZnCe sample exhibited only 1.4% of Con._{CO₂} at 280 °C. The CuZnCe sample only grew Ce-MOF with very low CuZn and showed poor performance. Thus, the three metals could not be added together bringing about the question of which element should be introduced first. When the Cu was introduced first, the CuZn-Ce sample with extremely low content of Zn (0.03%, Table 1) presented low performance as the Zn was lost during the second step of preparation. When the Zn was introduced first, the Zn-CuCe sample could not be prepared because that Zn-MOF was not generated under the similar conditions. When the Ce was introduced first, the CeCu-Zn with low Cu content and CeZn-Cu samples with low Zn content have bad performance because Ce-MOF could not be grown together with Cu/Zn-MOF. However, the Ce-CuZn sample could grow Ce-MOF and Cu/Zn-MOF well with two main steps, and thus it had high Cu/Zn-O_V-Ce species and exhibited the best performance. When the Zn content was increased, the Ce₂-CuZn₂ and Ce₁-CuZn₄ samples would decrease CO₂ conversion although the methanol yield was slightly increased. In addition, we also investigated more complicated preparation procedures with three main steps. It was shown that the Ce-Cu-Zn and Ce-Zn-Cu samples also have good performance, but the preparation procedures are more complicated. Furthermore, although the Sel._{MeOH} and STY_{MeOH} slightly increased over Ce₂-CuZn₂, Ce₁-CuZn₄ and Ce-Cu-Zn samples, they were increased at the expense of CO₂ conversion compared with Ce-CuZn sample. When the Con._{CO₂} over Ce-CuZn sample was also near 10%, the Sel._{MeOH} and STY_{MeOH} were increased to 45.5% and 154.0 g·kg_{cat}⁻¹·h⁻¹, respectively. Therefore, the optimized Ce-CuZn catalyst has the appropriate metal elements and suitable introduction order to form abundant Cu/Zn-O_V-Ce species, thus it presents the best catalytic performance for methanol synthesis compared with the control catalysts.”

Page 32: “It was found that the order of introduction of metal during MOFs preparation influences the growth of MOFs and hence the distribution of active Cu phases in the final catalyst.”

5. The DFT results were meaningful only under the condition that the model was built to match the real situation of the catalyst. This work must provide the basis for building the doping model, which can be explained by experiment of calculation.

Response: As the reviewer has pointed out, the DFT calculation models must be built to match the real situation of the catalyst in experiment, and the more explanation for the doping model should be provided. In fact, our experimental characterizations have provided the detailed evidence for the doping of Cu and Zn into CeO₂. As shown in **Figure 2c**, the Raman spectra results showed that the intensity of triply degenerate F_{2g} mode of CeO₂ became weaker and there was a distinct blue shift (466 to 461 cm⁻¹) from Ce-CuZn-IM to CuZn-Ce and Ce-CuZn samples, indicating that more copper species were doped into the ceria matrix. Meanwhile, an obvious broad band (550~650 cm⁻¹) induced

from the oxygen vacancies was also observed over CuZn-Ce and Ce-CuZn samples, which was attributed to the substitutional incorporation of Cu/Zn ions into the CeO₂ lattice. These results are also consistent with the EPR results, as illustrated in the above responses for the second and the third comments.

Figure 2c. Raman spectra of reduced CuZnCe samples.

Figure 2f. EPR spectra of the CuZnCe catalysts.

Moreover, the CO-DRIFTS (**Figure 5a**) and XPS (**Figure S12a** and **Table S1**) results confirm that the Ce-CuZn sample exhibits Cu⁰ and Cu⁺ species, in which the content of Cu⁺ species is up to 56.5%, indicating that abundant surface Cu⁺ species are responsible for the formation of Cu/Zn-O_v-

Ce active sites on the Ce-CuZn sample.

Furthermore, Cu K-edge EXAFS result (**Figure 3b**) showed that the Cu-O and Cu-Cu scattering path existed in the Ce-CuZn sample, Zn K-edge spectra (**Figure 3c**) showed that Ce-CuZn presents the higher oxidation state of Zn compared to ZnO, indicating that the doping of Zn into CeO₂ increased the electron density of Zn species. Thus, the Ce-CuZn sample existed in the form of part Cu and Zn species incorporated into CeO₂ lattice.

Figure 5a. CO-DRIFTS results of CuZnCe after purged by He for 20 min at 30 °C.

Figure S12. *Quasi in-situ* XPS profiles of the reduced CuZnCe catalysts. (a) Cu LMM, (b) O1s, (c) Ce3d.

Figure 3. XAS spectra of reduced Ce-CuZn and Ce-CuZn-IM samples. (a) Cu K-edge XANES and corresponding standard samples. (b) Fourier-transformed k^2 -weight EXAFS of Cu K-edge. (c) Zn K-edge XANES and corresponding standard samples. (d) Fourier-transformed k^2 -weight EXAFS of Zn K-edge. (e, f) The WT spectroscopy of Ce-CuZn sample.

For the construction of the models of Ce-CuZn sample, two neighboring Ce atoms on the $\text{CeO}_2(111)$ are replaced by two Cu atoms to reflect the Cu-Cu and Cu-O coordination structures. As shown in **Figure S19c**, the symmetrical oxygen atoms bonded to Cu atoms are removed to realize the charge balance, furthermore, the oxygen atom connecting two Cu atoms is removed to obtain the surficial Cu^+ . Taking the effect of Zn doping into account, a Ce atom on the $\text{CeO}_2(111)$ surface is replaced by a Zn atom, and an O atom bonded to Zn atom is removed to keep charge balance, named as $\text{Zn/Cu}^+-\text{CeO}_{2-x}$ (**Figure S19d**).

Figure S19. Surface morphology of (a) $\text{CeO}_2(\text{bulk})$, (b) $\text{O}_V\text{-CeO}_2(111)$, (c) $\text{Cu}^+\text{-CeO}_{2-x}(111)$ and (d) $\text{Zn/Cu}^+\text{-CeO}_{2-x}(111)$ surfaces.

Following the reviewer's valuable suggestions, more explanation for the selection of the doping model has been provided in the revised manuscript.

The addressed contents in the Results are as follows:

Page 26: "Combined with CO-DRIFTS and XPS results, it can be concluded that the Ce-CuZn sample exhibits both the Cu^0 and Cu^+ species, in which the content of Cu^+ species is up to 56.5%, indicating that abundant surface Cu^+ species are responsible for the formation of Cu/Zn- $\text{O}_V\text{-Ce}$ active sites on the Ce-CuZn sample. Meanwhile, the EXAFS result (Figure 3) showed that the Cu-O and Cu-Cu scattering path existed in the Ce-CuZn sample, thus, two Cu atoms are considered to replace the Ce atoms to reflect the Cu-Cu and Cu-O coordination structures. Firstly, we adopt the dominantly exposed (111) facet and further construct three models to stimulate CeO_2 sample with oxygen vacancy, Cu doped Ce-Cu sample and Cu/Zn co-doped Ce-CuZn sample (Figure S19), which are named as the $\text{O}_V\text{-CeO}_2$, $\text{Cu}^+\text{-CeO}_{2-x}$ and $\text{Zn/Cu}^+\text{-CeO}_{2-x}$, respectively. Given that Cu^+ acts as the main species in the Ce-CuZn sample, the differential charge density and Bader charge are employed to characterize the electronic effect of dopant Cu and Zn atoms."

The addressed contents in the Supplementary Information are as follows:

Page S13: "The defective $\text{CeO}_2(111)$ surface was modeled by removing an O atom adjacent to Ce atom, denoted as $\text{O}_V\text{-CeO}_2$ (Figure S19b). Two neighboring $\text{Ce}^{4+}\text{-O}^{2-}$ moiety on the $\text{CeO}_2(111)$ surface are replaced by two Cu^{2+} cations and two oxygen vacancies ($\text{Cu}^{2+}\text{-O}_V$). To obtain surficial Cu^+ , the oxygen atom connected with two Cu atoms is removed to balance the charge, named as $\text{Cu}^+\text{-CeO}_{2-x}$ (Figure S19c). Taking the effect of Zn doping into account, a Ce atom on $\text{CeO}_2(111)$ surface is replaced by a Zn atom and a O atom bonded to Zn atom are removed to keep charge balance, named as $\text{Zn/Cu}^+\text{-CeO}_{2-x}$ (Figure S19d)."

6. The performance data is really good from a pure perspective. However, one point to consider is that the content of Cu has exceeded 50%. So the economy and applicability of this system is to be examined. Secondly, the reaction conditions of this experiment are not exactly the same as those of other experiments, and it is meaningless to claim that it is STY maximum.

Response: It is significant to consider the economy and applicability of this system from an industrial application perspective. A recent article published in *Nature* shows that the cost of methanol made with CO₂ utilization is \$510/per tonne of methanol (**Table R1**)¹. The bulk (>85%) of the cost of CO₂-to-methanol production via hydrogenation is the capital and operating cost of H₂ production. Therefore catalyst improvements in terms of higher selectivity to methanol can result in higher efficiency of hydrogen utilization. The higher activity of the catalyst can enable operating at lower pressures and/or reducing the operating costs associated with recycling (and compressing) unconverted reactants.

Furthermore, we also have carefully considered the content of Cu over Ce-CuZn catalyst in this work has exceeded 50%. However, the content of Cu over the conventional CuZnAl catalysts for CO₂ hydrogenation to methanol is also usually in the range of 40~60% (**Table R2**). Moreover, the Pd loading over Pd-based catalysts for methanol synthesis would also be higher than 5% (**Table R2**), which is much more expensive than Cu-based catalysts. **Therefore, the Ce-CuZn catalyst with exceeded 50% of Cu would be still economical and applicable for CO₂ hydrogenation to methanol from the perspective of catalyst cost.**

To respond to *“Secondly, the reaction conditions of this experiment are not exactly the same as those of other experiments, and it is meaningless to claim that it is STY maximum”*, we agree with the review’s viewpoint and revised the expression as follows: “Compared with the state-of-the-art catalysts reported in the literature, **the Ce-CuZn sample also exhibits comparable STY of methanol under similar reaction conditions (Figure 4d and Table S6).**”

Table R1. Costs of utilization compared with product costs, scoping review¹.

Pathway	Cost of product made with CO ₂ utilization (US\$ per tonne of product) Median, scoping review	Selling price of product (US\$ per tonne of product) Present day	Difference (%)	Anticipated cost relative to incumbent in 2050 (summary, expert opinion survey and author group judgement)	Anticipated direction of cost relative to incumbent in 2050 (summary, expert opinion survey and author group judgement)
Polymers	1,440	2,040	-30%	Likely to be cheaper	Downward
Methanol	510	400	+30%	Insufficient consensus	Downward
Methane	1,740	360	+380%	Likely to be more expensive	Downward
Fischer-Tropsch fuels	4,160	1,200	+250%	Likely to be more expensive	Downward
Dimethyl ether	2,740	660	+320%	Insufficient consensus	Downward
Microalgae	2,680	1,000	+170%	Likely to be more expensive	Insufficient consensus
Aggregates	21	18	+20%	Insufficient consensus	Downward
Cement curing	56	71	-20%	Likely to be cheaper	Downward
CO ₂ -EOR	n.a.	n.a.	n.a.	Likely to be more expensive	Upward

Median cost estimates for products made with CO₂ utilization are derived from the backward-looking scoping review. References for the selling prices are set out in more detail in Supplementary Table 4. The costs and cost trends anticipated in 2050 are derived from a forward-looking expert opinion survey and from author group judgement.

Table R2. Typical Cu-based and Pd-based catalysts for CO₂ hydrogenation to methanol.

Catalysts	Metal loading	Ref.
Cu/ZnO/Al ₂ O ₃ -US	51% Cu	2
Cu/ZnO-CeO ₂	40.1% Cu	3
Cu/ZnO/Al ₂ O ₃	62.7% Cu	4
Cu/Zn/Al/Zr-F0	52.5% Cu	5
ZrO ₂ /Cu-0.1	90.3% Cu	6
CuZnAl-2-C	50.6% Cu	7
Pd/ZnO	5% Pd	8
Pd-Cu/SiO ₂	5.7% Pd	9-10

References

1. Hepburn, C. *et al.* The technological and economic prospects for CO₂ utilization and removal. *Nature* **575**, 87-97 (2019).
2. Dasireddy, V. D. B. C. & Likozar, B. The role of copper oxidation state in Cu/ZnO/Al₂O₃ catalysts in CO₂ hydrogenation and methanol productivity. *Renew. Energy* **140**, 452-460 (2019).
3. Zhu, J. *et al.* Flame synthesis of Cu/ZnO-CeO₂ catalysts: Synergistic metal-support interactions promote CH₃OH selectivity in CO₂ hydrogenation. *ACS Catal.* **11**, 4880-4892 (2021).
4. Kasatkin, I., Kurr, P., Kniep, B., Trunschke, A. & Schlogl, R. Role of lattice strain and defects in copper particles on the activity of Cu/ZnO/Al₂O₃ catalysts for methanol synthesis. *Angew. Chem. Int. Ed.* **46**, 7324-7327 (2007).
5. Gao, P. *et al.* Fluorinated Cu/Zn/Al/Zr hydrotalcites derived nanocatalysts for CO₂ hydrogenation to methanol. *J. CO₂ Util.* **16**, 32-41 (2016).
6. Wu, C. *et al.* Inverse ZrO₂/Cu as a highly efficient methanol synthesis catalyst from CO₂ hydrogenation. *Nat. Commun.* **11**, 5767 (2020).
7. Martin, O. *et al.* Zinc-rich copper catalysts promoted by gold for methanol synthesis. *ACS Catal.* **5**, 5607-5616 (2015).
8. Bahruji, H. *et al.* Pd/ZnO catalysts for direct CO₂ hydrogenation to methanol. *J. Catal.* **343**, 133-146 (2016).
9. Nie, X. *et al.* Mechanistic understanding of alloy effect and water promotion for Pd-Cu bimetallic catalysts in CO₂ hydrogenation to methanol. *ACS Catal.* **8**, 4873-4892 (2018).
10. Jiang, X., Koizumi, N., Guo, X. W. & Song, C. S. Bimetallic Pd-Cu catalysts for selective CO₂ hydrogenation to methanol. *Appl. Catal. B: Environ.* **170-171**, 173-185 (2015).

7. Further refinement of the manuscript graphics and tables are needed.

Response: We thank the reviewer's suggestions very much. We have carefully revised the graphics (**Figure 2, Figure 4, Figure 5, Figures S1, S2, S8-S12, S15-S17**) and tables (**Table 1, Tables S1, S2, S3, S4, S6, S7**) in the manuscript and supporting information. For example, the previous **Figure 2** and

Figure 4 have been optimized as follows:

Previous Figure 2 and Figure 4

Previous Figure 2. The crystalline phase and surface basicity of CuZnCe catalysts.

Revised Figure 2. The crystalline phase and surface basicity of CuZnCe catalysts.

Previous Figure 4. Catalytic performance for CO₂ hydrogenation over CuZnCe catalysts.

Revised Figure 4. Catalytic performance for CO₂ hydrogenation over CuZnCe catalysts.

Reviewer #2 (Remarks to the Author):

This work reports a high-performance Ce-CuZn catalyst with abundant Cu/Zn-O_v-Ce active sites for CO₂ hydrogenation to methanol. In recent years, the CO₂ hydrogenation as an effective method of CCUS is a very hot research topic that attract many readers. A robust Ce-CuZn catalyst in this study was achieved by a MOFs engineering method and the deep structure-performance relationship has been illustrated well. This is a systematic and innovative study work combined well with the DFT calculation. Many control experiments are carefully done and the conclusions are fully supported by the data presented. This work is meaningful to construct efficient multi-metal catalysts. Therefore, I would like to recommend publication of this manuscript in Nature Communications after addressing the following questions:

Response: We would like to thank the reviewer for his/her valuable comments.

1. It seems that the oxygen vacancy in the Cu/Zn-O_v-Ce active sites is a significant factor for methanol synthesis. How did the oxygen vacancies produce over the Ce-CuZn catalysts? The formation reason should be provided. The Raman spectra was used to study the oxygen vacancies in this work. In my opinion, more characterization experiments like the EPR should be done to confirm the oxygen vacancies.

Response: The oxygen vacancies over Ce-CuZn catalyst were produced from two aspects: I) generation from the reduction of Ce⁴⁺ to Ce³⁺, and II) the replacement of Ce⁴⁺ by Cu/Zn ions, resulting in the formation of oxygen vacancies¹.

We have done another two tests including electron paramagnetic resonance (EPR) and chemisorption measurements to confirm the oxygen vacancies. The chemisorption measurements results and analysis are illustrated in the response of the third question of **Reviewer #1**. The EPR results are depicted in **Figure 2f**. The EPR spectrum of Ce-CuZn sample presents obvious oxygen vacancies peak at g_{||} value of 1.99, which is much different with the other two samples². Moreover, the type K peaks at g_{||} values of 2.29 and 1.83 are ascribed to Cu²⁺/Zn²⁺ dimer, which could be observed when two neighboring Ce⁴⁺ ions with short separation distance are substituted by Cu²⁺/Zn²⁺ ions³. Thus, the appearance of K signals suggests that Cu/Zn-O_v-Ce solid solution is indeed generated in Ce-CuZn sample.

Figure 2f. EPR spectra of the CuZnCe catalysts.

References

1. Jiang, J., Yang, H., Jiang, H., Hu, Y. & Li, C. Boosting catalytic activity of Cu-Ce solid solution catalysts by flame spray pyrolysis with high Cu^+ concentration and oxygen vacancies. *Chem. Eng. J.* **471**, 144439 (2023).
2. Cheng, Q. *et al.* Amorphous/crystalline $\text{Cu}_{1.5}\text{Mn}_{1.5}\text{O}_4$ with rich oxygen vacancies for efficiently photothermocatalytic mineralization of toluene. *Chem. Eng. J.* **471**, 144295 (2023).
3. Chen, J. *et al.* Characterization and catalytic performance of Cu/CeO_2 and $\text{Cu}/\text{MgO}-\text{CeO}_2$ catalysts for NO reduction by CO. *Appl. Catal. A: Gen.* **363**, 208-215 (2009).

The added contents in the Results and Supplementary Information are as follows:

Page 15: “Moreover, EPR and chemisorption measurements were carried out to confirm the oxygen vacancies. The EPR spectrum of Ce-CuZn sample presents obvious oxygen vacancies peak at g_{\parallel} value of 1.99, which is much different with the other two samples (Figure S11). Moreover, the type K peaks at g_{\parallel} values of 2.29 and 1.83 are ascribed to $\text{Cu}^{2+}/\text{Zn}^{2+}$ dimer, which could be observed when two neighboring Ce^{4+} ions with short separation distance are substituted by $\text{Cu}^{2+}/\text{Zn}^{2+}$ ions. Thus, the appearance of K signals suggests that Cu/Zn-O_v-Ce solid solution is indeed generated in Ce-CuZn sample.”

Page S10: “**EPR test:** Electron paramagnetic resonance (EPR) measurements were performed at room temperature on a Bruker A 200 spectrometer. 10 mg sample was filled in the capillary tube for each measurement at room temperature. The detailed operation parameters were as below: The center field was 3320.00 G, the microwave

frequency was 9.3296 GHz, the modulation frequency was 100.00 kHz, and the conversion time was 82 ms. The g value was determined from precise frequency and magnetic field values.

2. In Table 1, it shows that the Cu loading is more than 50%, however, why the Cu lattice fringes was not observed in Figure 1c?

Response: Thank you for this question. The Cu lattice fringes was not observed mainly because that the lighter atomic weight of Cu with respect to Ce and the similar contrast of Cu and CeO₂ under electron beam, which was also observed in the other reported Cu/CeO₂ catalysts^{1,2}.

References

1. Wang, F. *et al.* A Photoactivated Cu-CeO₂ Catalyst with Cu-[O]-Ce Active Species Designed through MOF Crystal Engineering. *Angew. Chem. Int. Ed.* **59**, 8203-8209 (2020).
2. Zhang, Y. *et al.* Highly efficient Cu/CeO₂-hollow nanospheres catalyst for the reverse water-gas shift reaction: Investigation on the role of oxygen vacancies through in situ UV-Raman and DRIFTS. *Appl. Surf. Sci.* **516**, 146035 (2020).

The added contents in the Results are as follows:

Page 9: "The crystal phase of copper could not be observed due to the lighter atomic weight of Cu with respect to Ce and the similar contrast of Cu and CeO₂, which was observed in the other reported Cu/CeO₂ catalysts"

3. In Table 1, why did the control catalyst Ce-CuZn-IM exhibit much lower Cu loading and higher Zn loading than the Ce-CuZn catalyst? It suggests to control similar metal loading with Ce-CuZn catalyst and then to compare their catalytic performance. This control experiment should be added.

Response: The Ce-CuZn and Ce-CuZn-IM samples have the same amount of metal precursors. For the Ce-CuZn, some Zn ions can not be loaded into Ce-MOF and some Ce ions would be lost during centrifugation and washing procedures. For the Ce-CuZn-IM, the Ce-MOF was calcined to obtain the CeO₂ powder and then loaded with Cu and Zn without the centrifugation or washing procedures by the impregnation method. Therefore, the Ce-CuZn-IM exhibits much lower Cu loading and higher Zn loading than the Ce-CuZn catalyst.

We have prepared a Ce-CuZn-IM-B sample with a similar metal loading as the Ce-CuZn sample. But the Ce-CuZn-IM-B samples still showed much lower CO₂ conversion and STY of methanol compared with Ce-CuZn sample (**Table S4**).

Part of Table S4. Catalytic performance of the CuZnCe catalysts.

Catalysts	Con.	Sel.	STY _{CH₃OH} g·kg _{cat} ⁻¹ ·h ⁻¹	Reaction conditions
	CO ₂ %	CH ₃ OH %		
Ce-CuZn	15.1	26.4	140.6	280 °C, 2 MPa, 10,000 h ⁻¹ , H ₂ /CO ₂ /N ₂ =72/24/1
Ce-CuZn-IM	1.6	46.1	26.8	280 °C, 2 MPa, 10,000 h ⁻¹ , H ₂ /CO ₂ /N ₂ =72/24/1
Ce-CuZn-IM-B	3.0	42.6	56.7	280 °C, 2 MPa, 10,000 h ⁻¹ , H ₂ /CO ₂ /N ₂ =72/24/1

The addressed contents in the Results are as follows:

Page 10: “For the Ce-CuZn-IM sample, it has much lower Cu loading (22.36 wt.%) and more Zn loading (11.47 wt.%). This is because the Ce-MOF was calcined to obtain the CeO₂ powder and then loaded with Cu and Zn without the centrifugation or washing procedures by the impregnation method.”

Page 20: “As a result, the Ce-CuZn-IM-B samples still showed much lower CO₂ conversion and STY of methanol compared with Ce-CuZn sample (Table S4).”

4. A mistake is shown in the caption of Figure 2. There are two (c) captions in Figure 2. The H₂-TPR should be deleted as it has been shown in Figure S8.

Response: Thank you for pointing out this mistake. “(c) H₂-TPR curves.” has been deleted. In addition, the H₂-TPR curves have been moved into **Figure 2**.

The addressed contents in the Results are as follows:

Page 14: “**Figure 2. The crystalline phase and surface basicity of CuZnCe catalysts.** (a) Normal XRD patterns of the reduced samples tested offline. (b) *In-situ* XRD patterns of Ce-CuZn samples reduced at different temperature under the atmosphere of 40% H₂-N₂. (c) Raman spectra. (d) H₂-TPR curves. (e) CO₂-TPD curves. (f) EPR spectra.”

5. In Figure 2a, the diffraction peaks belong to Cu⁰ species is very obvious, however, why the CO adsorbed on Cu⁰ species were not be observed in Figure 5a?

Response: We have added the following explanation to our revised manuscript (Page 23):

The CO adsorbed on Cu⁰ species were not obvious at around 2058 cm⁻¹ for most of CuZnCe samples (**Figure 5a**), which was probably because that the *in-situ* CO-DRIFTS tests were operated at room temperature and the CO adsorption on Cu⁰ species were weak¹.

Figure 5a. CO-DRIFTS results of CuZnCe after purged by He for 20 min at 30 °C.

References

1. Chen, A. *et al.* Structure of the catalytically active copper-ceria interfacial perimeter. *Nat. Catal.* **2**, 334–341 (2019).

6. In Figure 4a, the CuZnCe, CuZn-Ce and Ce-CuZn samples with similar metals but presented distinct catalytic performance. The authors should provide more explanation to illustrate their difference.

Response: Yes, the three samples contained similar metals but presented distinct catalytic performances. For the CuZnCe sample, it exhibited poor catalytic performance with only 1.4% of $\text{Con.}_{\text{CO}_2}$ at 280 °C because the CuZnCe sample only grew Ce-MOF with very low CuZn metals and thus has bad performance. It suggests that the three metals could not be added together and need a step-by-step preparation. For the CuZn-Ce sample, it presented relatively low catalytic performance as the Zn was lost during the second step of preparation. However, the Ce-CuZn sample could grow Ce-MOF and Cu/Zn-MOF well with two main steps, and thus it has high Cu/Zn-O_v-Ce species and exhibits the best catalytic performance. More explanation is illustrated in the response of the 4th question of **Reviewer #1**.

7. Authors have used *in-situ* CO-DRIFTS to investigate the surface interactions with the CO

probe molecule, and further confirmed the existence of Cu^+ species over CuZnCe catalysts. Whether it can be verified the CO adsorption behaviors from DFT calculations?

Response: *Firstly*, according to the CO-DRIFTS results (**Figure 5a**), Ce-CuZn sample presents the strong peak at 2111 cm^{-1} that belongs to the adsorbed Cu^+ -CO species, and a weak peak at 2138 cm^{-1} that is belonged to the single-site $\text{Cu}^+/\text{Zn}-\text{O}_\text{V}-\text{Ce}$ species, confirming the existence of Cu^+ species on the Ce-CuZn sample. However, the adsorption configuration and energies of CO is still unclear. Thus, following the reviewer's valuable suggestion, the DFT calculation was employed to investigate the CO adsorption behavior.

Figure 5a. CO-DRIFTS results of CuZnCe after purged by He for 20 min at $30\text{ }^\circ\text{C}$.

Secondly, the supplemented CO adsorption behaviors over $\text{O}_\text{V}-\text{CeO}_2$, $\text{Cu}^+-\text{CeO}_{2-x}$, and $\text{Zn}/\text{Cu}^+-\text{CeO}_{2-x}$ catalysts have been systematically investigated using DFT calculations, as shown in **Table S7** and **Figure S23**.

Over $\text{O}_\text{V}-\text{CeO}_2$ catalyst, as shown in **Figure S23a**, CO can be adsorbed at the top of oxygen vacancy with the adsorption energy of $-36.7\text{ kJ}\cdot\text{mol}^{-1}$ at 0 K, and the C–O bond length is 1.141 \AA . The calculated result is consistent with the previous studies^{1,2}, which presents that the CO adsorption energy on the $\text{O}_\text{V}-\text{CeO}_2(111)$ is $-34.7\text{ kJ}\cdot\text{mol}^{-1}$ at 0 K, and the C–O bond length is 1.150 \AA .

Over $\text{Cu}^+-\text{CeO}_{2-x}$ catalyst, as shown in **Figure S23b**, CO is adsorbed at the Cu^+ top site with the

adsorption energy of $-61.3 \text{ kJ}\cdot\text{mol}^{-1}$ at 0 K, and the C–O bond is 1.167 \AA . Similarly, Wu *et al.* showed that CO molecule can be strongly bonded to Cu^+ species on the $\text{Cu}_m^+-\text{CeO}_x$ catalyst with the adsorption energy of $-55.96 \text{ kJ}\cdot\text{mol}^{-1}$ at 0 K³.

Over $\text{Zn}/\text{Cu}^+-\text{CeO}_{2-x}$, as shown in **Figure S23c**, CO is adsorbed at the Cu-Cu bridge site with the adsorption energy of $-87.5 \text{ kJ}\cdot\text{mol}^{-1}$ at 0 K, and the C–O bond length is 1.171 \AA .

The above results show that the adsorption and activation ability of CO over O_v-CeO_2 , $\text{Cu}^+-\text{CeO}_{2-x}$, and $\text{Zn}/\text{Cu}^+-\text{CeO}_{2-x}$ catalysts follow the order of $\text{Zn}/\text{Cu}^+-\text{CeO}_{2-x} > \text{Cu}^+-\text{CeO}_{2-x} > \text{O}_v-\text{CeO}_2$. Moreover, the most favorable adsorption site of CO is Cu^+ species on the $\text{Cu}^+-\text{CeO}_{2-x}$ and $\text{Zn}/\text{Cu}^+-\text{CeO}_{2-x}$ catalysts, which is consistent with the CO-DRIFTS results. Thus, both the CO-DRIFTS and DFT results for CO adsorption behaviors have confirmed the existence of Cu^+-CO species.

Figure S23. The optimized configurations of CO adsorption over (a) O_v-CeO_2 , (b) $\text{Cu}^+-\text{CeO}_{2-x}$, (c) $\text{Zn}/\text{Cu}^+-\text{CeO}_{2-x}$ catalysts.

Table S7. Adsorption free energies and key structural parameters of CO_2 , CO and H_2 species over O_v-CeO_2 , $\text{Cu}^+-\text{CeO}_{2-x}$ and $\text{Zn}/\text{Cu}^+-\text{CeO}_{2-x}$ catalysts at 553 (0) K.

Catalysts	Species	$G_{\text{ads}} (\text{kJ}\cdot\text{mol}^{-1})$	Adsorption site	$d_{\text{C-O}}/d_{\text{H-H}} (\text{\AA})$
O_v-CeO_2	CO_2 (linear)	-14.3 (-79.9)	O_v	1.167,1.186
	CO_2 (bent)	-30.6 (-105.8)	Ce-O bridge	1.269,1.268
	CO (mol)	37.7 (-36.7)	Ce-top	1.141
	H_2 (mol)	-19.3 (-70.1)	Ce-top	0.758
$\text{Cu}^+-\text{CeO}_{2-x}$	CO_2 (linear)	55.9 (-19.1)	O_v	1.170,1.181
	CO_2 (bent)	-48.2 (-124.1)	Ce-O bridge	1.289,1.264
	CO (mol)	23.8 (-61.3)	Cu-top	1.167
	H_2 (mol)	23.0 (-5.5)	Cu-Cu bridge	0.754
$\text{Zn}/\text{Cu}^+-\text{CeO}_{2-x}$	CO_2 (linear)	27.1 (-43.9)	O_v	1.172,1.181
	CO_2 (bent)	-58.5 (-140.4)	Ce-O bridge	1.252,1.316
	CO (mol)	18.3 (-87.5)	Cu-Cu bridge	1.171
	H_2 (mol)	1.7 (-7.4)	Cu-Cu bridge	0.754

*Parentheses: adsorption energies at 0 K.

References

1. Jiang, S. Y., Teng, B. T., Yuan, J. H., Guo, X. W., Luo, M. F. Adsorption and oxidation of CO over CeO₂(111) surface. *Acta Phys. -Chim. Sin.* **25**,1629-1634 (2009).
2. Guo, Y. *et al.* Low-temperature CO₂ methanation over CeO₂-supported Ru single atoms, nanoclusters, and nanoparticles competitively tuned by strong metal–support interactions and H-spillover effect. *ACS Catal.* **8**, 6203-6215 (2018).
3. Wu, D. C., Dong, K., Wu, D. Y., Fu, J. Y., Liu, H., Hu, S. W., Jiang, Z., Qiao, S. Z., Du, X. W. Cuprous ions embedded in ceria lattice for selective and stable electrochemical reduction of carbon dioxide to ethylene. *J. Mater. Chem. A* **6**, 9373 (2018).

Following the reviewer’s valuable suggestions, in the revised manuscript and **Supplementary Information**, the authors have added the following contents to describe CO adsorption behaviors based on DFT calculations.

The added contents in the Reaction mechanism and DFT calculations are as follows:

Page 29: “For CO adsorption, as presented in Figure S23 and Table S7, CO prefers to adsorb at the O_V site on the O_V-CeO₂, while the most favorable adsorption site over Cu⁺-CeO_{2-x} and Zn/Cu⁺-CeO_{2-x} catalysts is Cu⁺ site, which is consistent with the CO-DRIFTS results. Moreover, CO adsorption energies follow the order of Zn/Cu⁺-CeO_{2-x} (-87.5 kJ·mol⁻¹)>Cu⁺-CeO_{2-x} (-61.3 kJ·mol⁻¹)>O_V-CeO₂ (-34.7 kJ·mol⁻¹), and the C–O bond lengths also follow the same order of Zn/Cu⁺-CeO_{2-x} (1.171 Å)>Cu⁺-CeO_{2-x} (1.167 Å)>O_V-CeO₂ (1.141 Å), indicating that both the doping Cu and Cu/Zn promote the adsorption of CO molecule at the Cu⁺ site over Cu⁺-CeO_{2-x} and Zn/Cu⁺-CeO_{2-x} catalysts.”

The added contents in the revised Supplementary Information are as follows:

Page S34: “Over O_V-CeO₂, as shown in Figure S23a, CO can be adsorbed at the top of oxygen vacancy with the adsorption energy of -36.7 kJ·mol⁻¹ at 0 K, the C–O bond length is 1.141 Å. The calculated result is consistent with the previous studies,¹ which presents that the CO adsorption energy on the O_V-CeO₂(111) is -34.7 kJ·mol⁻¹ at 0 K, and the C–O bond length is 1.150 Å. Over Cu⁺-CeO_{2-x}, as shown in Figure S23b, CO is adsorbed at the Cu⁺ top site with the adsorption energy of -61.3 kJ·mol⁻¹ at 0 K, and the C–O bond is 1.167 Å. Similarly, Wu *et al.* showed that CO molecule can be strongly bonded to Cu⁺ species on the Cu^{m+}-CeO_x catalyst with the adsorption energy of -55.96 kJ·mol⁻¹ at 0 K. Over Zn/Cu⁺-CeO_{2-x}, as shown in Figure S21c, CO is adsorbed at the Cu-Cu bridge site with the adsorption energy of -87.5 kJ·mol⁻¹ at 0 K, and the C–O bond length is 1.171 Å.”

Thus, the adsorption and activation ability of CO over O_V-CeO₂, Cu⁺-CeO_{2-x}, and Zn/Cu⁺-CeO_{2-x} catalysts follow the order of Zn/Cu⁺-CeO_{2-x}>Cu⁺-CeO_{2-x}>O_V-CeO₂. Moreover, the most favorable adsorption site of CO is Cu⁺ species both on the Cu⁺-CeO_{2-x} and Zn/Cu⁺-CeO_{2-x} catalysts.”

8. The legend in Figure 6a seems wrong, only the differential charge density ($\Delta\rho$) of $\text{Cu}^+-\text{CeO}_{2-x}$ and $\text{Zn}/\text{Cu}^+-\text{CeO}_{2-x}$ catalysts are given in, while the $\text{Cu}^{2+}-\text{CeO}_{2-x}$ catalyst was missing.

Response: Authors are very sorry for the wrong expression in the legend of **Figure 6a**. Actually, the Cu^+ was the main species on the $\text{Cu}^+-\text{CeO}_{2-x}$ and $\text{Zn}/\text{Cu}^+-\text{CeO}_{2-x}$ catalysts, while $\text{Cu}^{2+}-\text{CeO}_{2-x}$ catalyst is not considered in this work. Thus, the electronic analysis for $\text{Cu}^{2+}-\text{CeO}_{2-x}$ is not given in **Figure 6a**.

Following the reviewer's valuable suggestions, in the revised manuscript, authors have corrected the legend of **Figure 6a**, and the corresponding contents have been readdressed.

The revised contents in the legend of Figure 6a are as follows:

Figure 6. DFT calculation results over O_v-CeO_2 , $\text{Cu}^+-\text{CeO}_{2-x}$ and $\text{Zn}/\text{Cu}^+-\text{CeO}_{2-x}$ catalysts to investigate the reaction mechanism. (a) Differential charge density ($\Delta\rho$) of $\text{Cu}^+-\text{CeO}_{2-x}$ and $\text{Zn}/\text{Cu}^+-\text{CeO}_{2-x}$ catalysts, the dark blue and light blue regions represent the charge accumulation and depletions.

9. Overall the manuscript is well written, could the authors provide more perspective on this research in the conclusion section?

Response: We have now expanded our perspective on this research in the **Discussion** section. As the **Reviewer #1** mentioned that the metal components should be optimized to consider its economy as the Cu loading has exceeded 50%. **Reviewer #3** has commented on the CO selectivity and the roles of oxygen vacancies. The methanol yield can be further improved as the CO selectivity is still high over CuZnCe catalysts. Thus, further research should be conducted to suppress the reverse water-gas shift reaction. In addition, as the polymetallic Cu/Zn- O_v -Ce active sites have presented excellent catalytic performance, the high-entropy alloy-based catalysts with various unique synergistic effects may have potential application in methanol synthesis, but it would be challenging to illustrate the polymetallic interfaces.

The added contents in the Discussion are as follows:

Page 33: “In addition, it is worth noting that oxygen vacancies in Cu/CeO₂ are not beneficial for CO hydrogenation because they are poisoned by adsorbed CO₂ to form of carbonate-like species. Thus, the roles of oxygen vacancies and their stability in CO/CO₂ hydrogenation still needs to be investigated in the development of new catalysts.”

Page 34: “In the future, the metal components should be optimized to consider its economy as the Cu loading has exceeded 50%. The methanol yield can be further improved as the CO selectivity is still high over CuZnCe catalysts. Thus, more strategies should be adopted to suppress the reverse water-gas shift reaction. In addition, as the polymetallic Cu/Zn-O_v-Ce active sites have presented excellent catalytic performance, the high-entropy alloy-based catalysts with various unique synergistic effects may have potential application in methanol synthesis, but it would be challenging to illustrate the polymetallic interfaces.”

Reviewer #3 (Remarks to the Author):

This manuscript, by Ye et al., studies the synthesis of CuZnCe catalysts using a MOF precursor. One of the catalysts shows a remarkable activity for methanol formation via CO₂ hydrogenation, comparable to technical CuZnAl catalysts.

Response: We would like to thank the reviewer for his/her valuable comments.

My general opinion is that in spite of all the characterizations performed, it is still not clear why the Ce-CuZn catalyst outperforms by large amount the other catalysts. The Zn-O_v-Ce sites that would explain its higher activity could also be present on the Ce-CuZn-IM. I think that more catalysts have to be prepared to assert which is the cause of the higher activity observed. The Zn content is too different (1% vs. 11%) and could have an effect. In fact, one important work that the authors do not discuss (Hensen et al, ACS Catal. 2021, 11, 4880–4892) shows that Zn content has a dramatic effect on CuZnCe catalysts for this reaction. I recommend to compare the results with that paper to enhance the discussion.

Response: We highly appreciate the reviewer's constructive suggestions to improve our manuscript. We have now prepared, characterized and tested many more catalysts (e.g., CeCu-Zn, CeZn-Cu, Zn-CuCe, Ce₂-CuZn₂, Ce₁-CuZn₄, Ce-Cu-Zn, Table S4) and report the results in our revised manuscript. We have performed new characterizations (TG, EPR, *quasi in-situ* XPS test, TEM, chemisorption measurements, CO-DRIFTS, and transient *in-situ* DRIFTS) to support our conclusions. Some significant results are depicted below:

1. The effect of Zn content on the catalytic performance (Table S4).

When the Ce-Cu catalyst without Zn promoter, its methanol yield was much lower than that over Ce-CuZn. When the Zn content was increased, the Ce₂-CuZn₂ and Ce₁-CuZn₄ samples showed decreased CO₂ conversion although the methanol yield was slightly higher than that over Ce-CuZn. When the Con._{CO₂} over Ce-CuZn sample was also near 10%, the Sel._{MeOH} and STY_{MeOH} were increased to 45.5% and 154.0 g·kg_{cat}⁻¹·h⁻¹, respectively.

As the reviewer has pointed out, the Zn content has a dramatic effect on CuZnCe catalysts for CO₂ hydrogenation to CH₃OH, and the results in this work should be compared with the reference (Hensen et al., ACS Catal. 2021, 11, 4880-4892) to enhance the discussion. A similar Zn promotion effect on CO₂ hydrogenation to CH₃OH has been reported,¹ and the addition of Zn in the Cu-Zn-Ce

oxide catalysts was beneficial to decorate Cu active sites and thus inhibit the RWGS reaction by Cu-CeO₂ interactions, as a result, CO₂ hydrogenation to CH₃OH preferred to occur by the formate pathway, and CH₃OH selectivity as well as CO₂ conversion can be improved (**Figure R1**). Similarly to the literature report, the addition of small amount of Zn into Ce-Cu catalyst could enhance both CO₂ conversion and methanol selectivity due to the inhibition of RWGS reaction. However, when Zn content increased much, the Zn²⁺ and Cu²⁺ were competitively grown into Ce-MOF, which would affect the Cu loading over the Ce₂-CuZn₂ and Ce₁-CuZn₄ samples. Thus, the CO₂ conversion was decreased and the methanol selectivity increased much.

Figure R1. (a) CO₂ conversion and CH₃OH selectivity; (b) CH₃OH and CO formation rates as a function of support composition for Cu(45)/ZnO-CeO₂(y) catalysts. Reaction conditions: 250 °C, 30 bar, H₂/CO₂/N₂ = 3:1:1, and SV = 120 L/(g_{cat} × h).

2. The effect of addition order of Zn on the catalytic performance (Table S4).

When the Zn was introduced first, the Zn-CuCe sample was even not prepared because that Zn-MOF could not be generated under the similar conditions. Moreover, both CeZn-Cu and CeCu-Zn samples presented much lower catalytic performance than Ce-CuZn due to CeZn-MOF and CeCu-MOF could not be grown together. In addition, we also investigated more complicated preparation procedures with three main steps. It was shown that the Ce-Cu-Zn and Ce-Zn-Cu samples also have good performance, but the preparation procedures are more complicated.

3. Why the Ce-CuZn catalyst outperforms by large amount the other catalysts?

The Ce-CuZn catalyst from atomic-level substitution of Cu and Zn into Ce-MOF precursor

produced many active Cu/Zn-O_V-Ce species. The synergistic effect of CuZnCe in the Cu/Zn-O_V-Ce species catalyses CO₂ activation and the hydrogenation of formates to achieve a high yield of methanol. Moreover, the Ce-CuZn catalyst has abundant Cu⁺ species and the highest amount of oxygen vacancies. Kumari *et al.* reported that the number of oxygen vacancies could influence CO₂ activation via lower reaction barriers of CO₂ dissociation². Our DFT calculation results have shown that compared to O_V-CeO_{2-x} catalyst, the Cu⁺-CeO_{2-x} catalyst with Cu doping into CeO_{2-x} reduces the activation barrier of CO₂ hydrogenation to generate *HCOO. Furthermore, the incorporation of Cu and Zn into CeO₂ with abundant oxygen vacancies can facilitate the H₂ dissociation and the formation of *HCOO, thus improving CO₂ hydrogenation over Ce-CuZn catalyst *via* formate intermediates. In addition, the Ce-CuZn sample have more weaker basic sites for CO₂ adsorption.

In summary, the optimized Ce-CuZn catalyst has the appropriate metal elements and suitable introduction order to form abundant Cu/Zn-O_V-Ce species, thus it presents the best catalytic performance for methanol synthesis compared with the control catalysts.

Part of Table S4. Catalytic performance of the CuZnCe catalysts. ^{a)}

Catalysts	Con. CO ₂ %	Sel. CH ₃ OH %	STY _{CH₃OH} g·kg _{cat} ⁻¹ ·h ⁻¹	Remark
Ce	0.38	0	0	The CeO ₂ support without Cu or Zn has no activity.
CuZn	0.9	38.4	11.7	CuZn without CeO ₂ support would be sintering and has poor activity.
Ce-Cu ^{b)}	9.7	16.9	58.1	The sample without Zn promoter has low performance.
Ce-CuZn	15.1	26.4	140.6	The sample has high Cu/Zn-O_V-Ce species and exhibits the best performance.
CeCu-Zn	1.05	34.2	12.7	The sample with low Cu content has bad performance due to CeCu-MOF could not be grown together.
CeZn-Cu	5.3	20.5	38.6	The sample with low Zn content has bad performance due to CeZn-MOF could not be grown together.
Zn-CuCe	NA	NA	NA	The sample could not be prepared due to Zn-MOF could not be prepared under the similar conditions.
Ce ₂ -CuZn ₂	10.9	38.4	148.6	The samples with increased Zn content would decrease CO ₂
Ce ₁ -CuZn ₄	8.9	47.4	149.6	conversion although the methanol yield was slightly increased.
Ce-Cu-Zn ^{d)}	10.7	43.3	165.7	The two samples also have good performance, but the

Ce-Zn-Cu ^{d)}	12.4	23.9	106.2	preparation procedures are more complicated.
Ce-CuZn-IM	1.6	46.1	26.8	The sample prepared with impregnation method (IM) has lower performance than Ce-CuZn.
Ce-CuZn-IM-B	3.0	42.6	56.7	The sample with similar Cu and Zn loading but still has lower performance than Ce-CuZn.

a) Reaction conditions: 280 °C, 2 MPa, 10,000 h⁻¹, H₂/CO₂/N₂=72/24/1.

b) “-” means two main steps for the sample preparation. The Ce-MOF was grown first and then the Cu-MOF was grown on the Ce-MOF. It is the same for other samples.

c) Reaction temperature for this sample was 320 °C due to its extremely low performance.

d) “- -” means three main steps for the sample preparation.

Reference

- Zhang, J. D. *et al.* Flame synthesis of Cu/ZnO-CeO₂ catalysts: Synergistic metal-support interactions promote CH₃OH selectivity in CO₂ hydrogenation. *ACS Catal.* **11**, 4880-489 (2021).
- Kumari, N., Haider, M. A., Agarwal, M., Sinha, N., & Basu, S. Role of reduced CeO₂(110) surface for CO₂ reduction to CO and methanol. *J. Phys. Chem. C* **120**, 16626–16635 (2016).

The addressed contents in the Discussions are as follows:

Page 18: “To assess the effect on performance of the order in which the ions are introduced, more control samples have been prepared and evaluated, as illustrated in Table S4. The pure CeO₂ support and the binary system of CuZn, Ce-Cu, and Ce-Zn catalysts have presented poor catalytic performance. Thus, it is necessary to prepare the ternary system. For the order of introducing the Cu, Zn, and Ce elements, they were firstly introduced together, but the CuZnCe sample exhibited only 1.4% of Con._{CO2} at 280 °C. The CuZnCe sample only grew Ce-MOF with very low CuZn and showed poor performance. Thus, the three metals could not be added together bringing about the question of which element should be introduced first. When the Cu was introduced first, the CuZn-Ce sample with extremely low content of Zn (0.03%, Table 1) presented low performance as the Zn was lost during the second step of preparation. When the Zn was introduced first, the Zn-CuCe sample could not be prepared because that Zn-MOF was not generated under the similar conditions. When the Ce was introduced first, the CeCu-Zn with low Cu content and CeZn-Cu samples with low Zn content have bad performance because Ce-MOF could not be grown together with Cu/Zn-MOF. However, the Ce-CuZn sample could grow Ce-MOF and Cu/Zn-MOF well with two main steps, and thus it had high Cu/Zn-O_v-Ce species and exhibited the best performance. When the Zn content was increased, the Ce₂-CuZn₂ and Ce₁-CuZn₄ samples would decrease CO₂ conversion although the methanol yield was slightly increased. In addition, we also investigated more complicated preparation procedures with three main steps. It was shown that the Ce-Cu-Zn and Ce-Zn-Cu samples also have good performance, but the preparation procedures are more complicated. Furthermore, although the Sel._{MeOH} and STY_{MeOH} slightly increased over Ce₂-CuZn₂, Ce₁-CuZn₄ and Ce-Cu-Zn samples, they were increased at the expense of CO₂ conversion compared with Ce-CuZn sample. When the Con._{CO2} over Ce-CuZn sample was also near 10%, the Sel._{MeOH} and STY_{MeOH} were increased to 45.5%

and $154.0 \text{ g} \cdot \text{kg}_{\text{cat}}^{-1} \cdot \text{h}^{-1}$, respectively. Therefore, the optimized Ce-CuZn catalyst has the appropriate metal elements and suitable introduction order to form abundant Cu/Zn-O_V-Ce species, thus it presents the best catalytic performance for methanol synthesis compared with the control catalysts.”

Page 32: “Compared to O_V-CeO_{2-x} catalyst, the Cu⁺-CeO_{2-x} catalyst with Cu doping into CeO_{2-x} reduces the activation barrier of CO₂ hydrogenation to generate *HCOO. Furthermore, doping Zn into Cu⁺/CeO_{2-x} can largely facilitate H₂ dissociation and the formation of *HCOO. A similar Zn promotion effect on CO₂ hydrogenation to CH₃OH has already been reported, and the addition of Zn in the Cu-Zn-Ce oxide catalysts was beneficial to inhibit the RWGS reaction. As a result, CO₂ hydrogenation to methanol is preferred to occur by the formate pathway, and CH₃OH selectivity can be improved.”

Page 33: “In summary, the Ce-CuZn catalyst outperformed by large amount the other catalysts is attributed to the following reasons: The Ce-CuZn catalyst from atomic-level substitution of Cu and Zn into Ce-MOF precursor produced many active Cu/Zn-O_V-Ce species. Moreover, the Ce-CuZn catalyst has abundant Cu⁺ species, large amount of oxygen vacancies and more weaker basic sites for CO₂ adsorption. Kumari *et al.* reported that the number of oxygen vacancies could influence CO₂ activation via lower reaction barriers of CO₂ dissociation. Our DFT calculation results have shown that the incorporation of Cu and Zn into CeO₂ with abundant oxygen vacancies can facilitate the H₂ dissociation and the formation of *HCOO, thus improving CO₂ hydrogenation over Ce-CuZn catalyst *via* formate intermediates.”

Authors claim that the peak at 183 °C observed in the TPR curves indicates the interaction of Cu-Zn and Cu-Ce, which is related to oxygen vacancies. If this is correct, the Ce-CuZn-IM would also have an important amount of Zn-O_V-Ce sites. Since it contains more Zn than Ce-CuZn, Ce-CuZn-IM would also have more vacancies related to Zn. How this correlates with the poor activity of Ce-CuZn-IM catalyst?

Response: The authors have not related the peak at 183 °C to oxygen vacancies. To solve this great question, the H₂-TPR profiles were fitted with three peaks (**Figure S10**), which were attributed to the reduction of dispersed copper species that weakly interact with CeO₂ (peak α), bulk CuO and dispersed copper species that strongly interact with CeO₂ (peak β), and Cu/Zn-O_V-Ce solid solution (peak γ)^{1,2}. It shows that Ce-CuZn and Ce-CuZn-IM samples exhibit higher reduction temperatures than that of CuZn-Ce, which is probably because that the former two samples have higher contents of Zn to decorate the Cu particles². Moreover, the order of peak γ ratio over CuZnCe catalysts is as follows: Ce-CuZn > CuZn-Ce > Ce-CuZn-IM, indicating the existence of many Cu/Zn-O_V-Ce species with strong metal-support interaction over the Ce-CuZn sample. The above characterization results

indicate that the Ce-CuZn sample exhibits Cu^0 , Cu_2O and $\text{Cu}^+/\text{Zn-O}_\text{v}\text{-Ce}$ species.

The main peaks at 167, 180, 183 °C are assigned to bulk CuO and dispersed copper species that strongly interact with CeO_2 (peak β), which is not related with the oxygen vacancies. Furthermore, although the Ce-CuZn-IM contains more Zn than Ce-CuZn, the Ce-CuZn-IM would also not have more vacancies related to Zn due to the different introduction method for Zn species. The Ce-CuZn sample was prepared by atomic-level substitution of Zn into Ce-MOF precursor while the latter sample was prepared by impregnation method. Therefore, the Ce-CuZn-IM sample exhibits fewer oxygen vacancies, which is demonstrated by Raman, EPR, and chemisorption results (please check the response of the third question of **Reviewer #1** for further details on these characterization results).

Figure 2d and Figure S10. H_2 -TPR curves of CuZnCe catalysts with corresponding fitting results.

References

- Zhu, J. *et al.* Flame synthesis of Cu/ZnO-CeO₂ catalysts: Synergistic metal-support interactions promote CH₃OH selectivity in CO₂ hydrogenation. *ACS Catal.* **11**, 4880-4892 (2021).
- Zhu, J. D. *et al.* Mechanism and nature of active sites for Methanol Synthesis from CO/CO₂ on Cu/CeO₂. *ACS*

The added contents in the Results are as follows:

Page 12: “The temperature-programmed reduction (H₂-TPR) profiles were fitted with three peaks (Figure 2d and Figure S10), which were attributed to the reduction of dispersed copper species that weakly interact with CeO₂ (peak α), bulk CuO and dispersed copper species that strongly interact with CeO₂ (peak β), and Cu/Zn-O_V-Ce solid solution (peak γ). It shows that Ce-CuZn and Ce-CuZn-IM samples exhibit higher reduction temperatures than that of CuZn-Ce, which is probably because that the former two samples have higher contents of Zn to decorate the Cu particles. Moreover, the order of peak γ ratio over CuZnCe catalysts is as follows: Ce-CuZn > CuZn-Ce > Ce-CuZn-IM, indicating the existence of many Cu/Zn-O_V-Ce species with strong metal-support interaction over the Ce-CuZn sample. The above characterization results indicate that the Ce-CuZn sample exhibits Cu⁰, Cu₂O and Cu⁺/Zn-O_V-Ce species.”

XAS analysis. The main finding is that, for Ce-CuZn, “... we speculate that some Cu and Zn were incorporated into CeO₂ lattice to form Cu/Zn-O_V-Ce species while the other Cu and Zn on the catalyst surface combined with CeO₂ to form Cu/Zn-ceria interfaces...” This result could be valid also for Ce-CuZn-IM? For this catalyst, the fact that there is more Zn (11% vs. 1% for Ce-CuZn) could facilitate the formation of CuZn alloy but also a part of Ce could be forming the Cu/Zn-O_V-Ce species.

Response: We highly appreciate the reviewer very much for the careful reading. According to the ICP results in **Table 1**, the Ce-CuZn sample possessed more copper content (~ 52.82 wt%) and lower Zn loading (~ 1.12 wt%), compared to that of the Ce-CuZn-IM sample which contained less copper (22.36%) and more zinc (11.47%), attributed to the different preparation methods. Although the Ce-CuZn-IM contains more Zn than Ce-CuZn, the Ce-CuZn-IM would also not have more Cu/Zn-O_V-Ce species due to the different introduction method for Zn species. The Ce-CuZn sample was prepared by atomic-level substitution of Zn into Ce-MOF precursor while the latter sample was prepared by impregnation method. Therefore, the Ce-CuZn-IM sample exhibits less Cu/Zn-O_V-Ce species than those over Ce-CuZn sample, which is also demonstrated by Raman, EPR, XPS, and chemisorption results (please refer to the response of the third question of **Reviewer #1** for details). In particular, the XPS results of the Ce-CuZn-IM suggest that Zn species prefer to move to the catalyst surface (**Figure S11**). Thus, the Zn species over the Ce-CuZn-IM samples are mainly ZnO, and also a part of CuZn alloy and Cu/Zn-O_V-Ce species.

Figure S11. *Quasi in-situ* XPS profiles of the reduced CuZnCe catalysts: (a) Cu2p, (b) Zn2p.

The added contents in the Results are as follows:

Page 17: “Since the higher oxidation state of Zn in Ce-CuZn, and almost no CuZn alloy generated in the catalyst, we speculate that some Cu and Zn were incorporated into CeO₂ lattice to form Cu/Zn-O_v-Ce species while the other Cu and Zn on the catalyst surface combined with CeO₂ to form Cu/Zn-ceria interfaces. While the Zn species over the Ce-CuZn-IM samples are mainly ZnO, and also a part of CuZn alloy and Cu/Zn-O_v-Ce species.”

CO adsorption monitored by IR. In line 390: “...In addition, the Ce-CuZn and CuZn-Ce samples present an obvious shoulder peak at 2138 cm⁻¹, which belongs to single-site Cu⁺/Zn-O-Ce species...” How this is possible if CuZn-Ce has a very tiny amount of Zn? Please discuss it. It is possible that this peak is related only to Cu⁺-O_v-Ce³⁺ vacancies?

Response: Thank you for this interesting question. To verify the origin of the shoulder peak at 2138 cm⁻¹, we have supplemented the CO-DRIFTS spectra over Ce-Cu and Ce-Zn samples (**Figure 5a**). The control samples, especially Ce-Cu sample, did not show the shoulder peak at 2138 cm⁻¹. Thus, it indicates that the shoulder peak at 2138 cm⁻¹ belongs to monocarbonyl adsorbed on single-site Cu⁺/Zn-O_v-Ce species located in a constrained environment. The Zn content (1.12%) was indeed low but Zn was highly dispersed from the Zn mapping image (**Figure 1e**). Moreover, the DRIFTS is an extremely sensitive instrument.

“It is possible that this peak is related only to Cu⁺-O_v-Ce³⁺ vacancies?” We think it is possible. The CuZn-Ce sample exhibited only a tiny amount of Zn content (0.03%) and the Ce-CuZn-IM exhibited high Zn content (11.47%), but the former one still presented the shoulder peak at 2138 cm⁻¹

¹ and the latter one did not show this shoulder peak. Also, the literatures report that similar shoulder peaks at 2135 and 2136 cm^{-1} were observed over the Cu-SSZ-13 zeolite catalyst and Cu/UiO-66 single-atom catalyst, respectively^{1,2}. These peaks are attributed to CO molecule adsorbed onto Cu^+ ions located in constrained environments of zeolites or MOFs. Thus, the shoulder peak at around 2138 cm^{-1} was related to the confined Cu^+ microenvironments instead of Zn contents.

Figure 5. (a) CO-DRIFTS results of CuZnCe after purged by He for 20 min at 30 °C.

Figure 1. (d-g) The TEM and HRTEM images of reduced Ce-CuZn sample with corresponding elemental mapping.

References

1. Abdel-Mageed, A. M. *et al.* Highly active and stable single-atom Cu catalysts supported by a metal-organic framework. *J. Am. Chem. Soc.* **141**, 5201-5210 (2019).
2. Szanyi, J., Kwak, J. H., Zhu, H. & Peden, C. H. Characterization of Cu-SSZ-13 NH_3 SCR catalysts: an in situ FTIR

The addressed contents in the Results are as follows:

Page 23 "In addition, the Ce-CuZn and CuZn-Ce samples present an obvious shoulder peak at 2138 cm^{-1} , which belongs to single-site Cu^+ ions located in constrained environments."

How is the selectivity to CO of Ce-CuZn compared to technical CuZnAl?

Response: As shown in Figure 4c, the CO selectivity of Ce-CuZn was also stable and the average value was 26.7%. Compared to some technical CuZnAl catalysts, the Ce-CuZn have presented lower CO selectivity (**Table S6**). However, future research should also focus on how to inhibit the RWGS reaction and decrease the CO selectivity.

Figure 4c. Catalytic stability of Ce-CuZn catalyst. Reaction conditions: GHSV=20,000 $\text{mL}/(\text{g}_{\text{cat}}\cdot\text{h})$, P= 2.8 MPa, T= 260 °C, H_2 : CO_2 : N_2 = 72: 24: 1.

Table S6. Representative catalysts for CO_2 hydrogenation to methanol.

Catalysts	Con. ^a %	Sel. ^b %	STY ^c	T, °C	P, MPa	GHSV ^d	H_2/CO_2 ratio	Ref.
Cu/ZnO/Al ₂ O ₃ -1	23.1	31.2/NA ^e	123.7 ^g	260	3	6,000	3	1
Cu/ZnO/Al ₂ O ₃ -2	18.2	41.1/NA ^e	410	260	5	15,000	3	2
Cu/ZnO/Al ₂ O ₃ -3	19.7	39.7/59.7 ^e	340	250	5	12,000	3	3

Cu/ZnO/Al ₂ O ₃ -4	11.1	54.8/45.2 ^e	391.3 ^g	250	4	18,000	3	4
Cu/ZnO/Al ₂ O ₃ -5	13.4	58.1/41.6 ^e	250	230	5	10,000	3	5
Cu/ZnO/Al ₂ O ₃ -6	24.6	67.1/32.9 ^e	283.2 ^g	260	5	5,000	3	6
Ce-CuZn-MOF	8.0	71.1/26.7 ^e	400.3	260	2.8	20,000	3	h

^a CO₂ conversion. ^b Selectivity of methanol. ^c Space time yield of methanol (g·kg_{cat}⁻¹·h⁻¹). ^d Gas hourly space velocity (mL·g_{cat}⁻¹·h⁻¹). ^e The former data is the selectivity of methanol while the latter one is the selectivity of CO, and NA is the abbreviation of not available. ^f The data was read from the figure in the original references. ^g Calculated according to the data. ^h This work.

References

1. Tan, Q., Shi, Z. & Wu, D. CO₂ hydrogenation to methanol over a highly active Cu–Ni/CeO₂-nanotube catalyst. *Ind. Eng. Chem. Res.* **57**, 10148-10158 (2018).
2. Hu, J. *et al.* Sulfur vacancy-rich MoS₂ as a catalyst for the hydrogenation of CO₂ to methanol. *Nat. Catal.* **4**, 242-250 (2021).
3. Gao, P. *et al.* Influence of modifier (Mn, La, Ce, Zr and Y) on the performance of Cu/Zn/Al catalysts via hydrotalcite-like precursors for CO₂ hydrogenation to methanol. *Appl. Catal. A: Gen.* **468**, 442-452 (2013).
4. An, B. *et al.* Confinement of ultrasmall Cu/ZnO_x nanoparticles in metal-organic frameworks for selective methanol synthesis from catalytic hydrogenation of CO₂. *J. Am. Chem. Soc.* **139**, 3834-3840 (2017).
5. Gao, P. *et al.* Yttrium oxide modified Cu/ZnO/Al₂O₃ catalysts via hydrotalcite-like precursors for CO₂ hydrogenation to methanol. *Catal. Sci. Technol.* **5**, 4365-4377 (2015).
6. Wang, X. *et al.* Catalytic activity for direct CO₂ hydrogenation to dimethyl ether with different proximity of bifunctional Cu-ZnO-Al₂O₃ and ferrierite. *Appl. Catal. B: Environ.* **327**, 122456 (2023).

The addressed contents in the Results are as follows:

Page 20: “The average Con._{CO₂}, Sel._{MeOH}, Sel._{CO}, and STY of methanol over Ce-CuZn sample during time on stream of 170 h were 8.0%, 71.1%, 26.7%, and 400.3 g·kg_{cat}⁻¹·h⁻¹, respectively (Figure 4c), which were stable without obvious decrease. Compared to some technical CuZnAl catalysts, the Ce-CuZn have presented lower CO selectivity (Table S6).”

Why is the data for CuZn missing from table 1?

Response: The CuZn is a control sample, and now the missed data has been added. The results show that the S_{BET}, S_{Cu}, and D_{Cu} of CuZn sample are much lower than the other CuZnCe samples. Thus we loaded CuZn to CeO₂ support. However, two missed data for CuZn are because CuZn does not contain Ce and the oxygen vacancies are from CeO₂ support.

The added contents in the Results are as follows:

Page 11: Table 1. The physicochemical properties of CuZnCe catalysts.

Catalysts	Cu ^{a)} (wt.%)	Zn ^{a)} (wt.%)	Ce ^{a)} (wt.%)	S _{BET} (m ² /g)	Pore size (nm)	Pore volume (cm ³ /g)	S _{Cu} ^{b)} (m ² /g)	D _{Cu} ^{b)} (%)	N _{OV} (μ mol/g _{cat}) ^{b)}
CuZn	80.65	3.55	—	2.9	26.9	0.01	3.7	0.7	—
CuZn-Ce	52.33	0.03	28.71	41.5	6.7	0.07	31.1	9.2	6.0
Ce-CuZn	52.82	1.12	31.74	40.2	12.1	0.12	23.1	6.7	18.1
Ce-CuZn-IM	22.36	11.47	48.80	41.4	7.2	0.08	22.0	15.2	2.9

^{a)} Metal loading results from ICP.

^{b)} Metallic copper surface area (S_{Cu}), copper dispersion (D_{Cu}) and oxygen vacancies (N_{OV}) determined by N₂O titration and H₂ temperature-programmed reduction (H₂-TPR).

In situ DRIFTS analysis at steady state show the typical surface species that previous authors have shown in these type of catalysts. The presence of these surface species is not a direct evidence of a determined pathway for methanol formation. More experiments are required (transient for example) to assess the role of each surface species in the mechanism and discard spectators.

Response: Thank you for the suggestion and we have now conducted transient *in-situ* DRIFTS experiments to gain more information about the reactivity of surface species. The Ce-CuZn catalyst was firstly exposed to pure CO₂ then the system was switched to other reaction atmospheres, resulting in significant change of the surface species, as shown in **Figure 5c** and **Figure S18**. Firstly, when the CO₂ gas was injected to the system, different types of hydroxyl groups at around 3730~3602 cm⁻¹ appeared and increased with time. Some other carbonates (1522, 1330 cm⁻¹) and formate (1585, 1374 cm⁻¹) peaks are also shown in **Figure S18a**. When the gas was switched to H₂, the above peaks became weak and peaks of hydroxyl groups disappeared (**Figure S18b**). In particular, the intensity of formate at 1585 cm⁻¹ increased firstly and then decreased slightly when the system was dosed CO₂ (Figure 5c), which was because that CO₂ firstly reacted with hydrogen available on the catalyst surface to generate formate but then decreased due to conversion to methoxy and lack of hydrogen. Thus, when the hydrogen was injected into the system, the intensity of formate increased quickly and then decreased with time due to consumption of adsorbed CO₂ and conversion of formate to methanol.

Secondly, upon switching the reaction atmosphere from hydrogen to CO₂ + H₂, the OH* peaks became positive but then became negative after 120 min (**Figure S18c**), which was much different with the pure CO₂ atmosphere. It was possible that the surface hydroxyl groups reacted with CO₂ and the bicarbonates were converted to methanol. The formate peaks were also decreased first and then increased to a stable state during reaction. **Finally**, the gas flow was switched to Ar, and the OH* peaks disappeared and other species peaks became weaker and weaker. However, the formate and carbonates could still be observed after purging for 30 min (**Figure S18d**), indicating that these surface species were stable. In addition, the CO* peaks at 2178 and 2110 cm⁻¹ could not be observed during the transient *in-situ* DRIFTS experiments, thus these CO* species were regarded mainly as spectators during the reaction. Therefore, the surface species of formate, carbonates, bicarbonate, and methoxy are proposed as the main reaction intermediates in the mechanism of CO₂ hydrogenation to methanol.

Figure S18. Transient *in-situ* DRIFTS experiments on Ce-CuZn catalyst collected at 260 °C when the reaction atmospheres were switched from (a) CO₂ to (b) H₂, (c) CO₂+H₂, and (d) Ar.

Figure 5c. Transient *in-situ* DRIFTS experiments on Ce-CuZn sample at 260 °C, and the peak intensity of HCOO* species at 1585 cm⁻¹ as a function of time when the reaction atmospheres were switched from CO₂ to H₂, CO₂+H₂, and Ar.

The addressed contents in the Results are as follows:

Page 24: “Moreover, the transient *in-situ* DRIFTS experiments were performed to demonstrate the formate mechanism. The Ce-CuZn catalyst was firstly exposed to pure CO₂ then the system was switched to other reaction atmospheres (H₂, CO₂+H₂, Ar), resulting in a significant change of the surface species, as shown in Figure 5c and Figure S18. Firstly, when the CO₂ gas was injected to the system, different types of hydroxyl groups at around 3730~3602 cm⁻¹ appeared and increased with the time. Some other carbonates (1522, 1330 cm⁻¹) and formate (1585, 1374 cm⁻¹) peaks are also shown in Figure S18a. When the gas was switched to H₂, the above peaks became weak and peaks of hydroxyl groups disappeared (Figure S18b). In particular, the intensity of formate at 1585 cm⁻¹ increased firstly and then decreased slightly when the system was dosed CO₂ (Figure 5c), which was because that CO₂ firstly reacted with hydrogen available on the catalyst surface to generate formate but then decreased due to conversion to methoxy and lack of hydrogen. Thus, when the hydrogen was injected into the system, the intensity of formate increased quickly and then decreased with time.

Secondly, upon switching the reaction atmosphere from hydrogen to CO₂ + H₂, the OH* peaks became positive but then became negative after 120 min (Figure S18c), which was much different with the pure CO₂ atmosphere. It was possible that the surface hydroxyl groups reacted with CO₂ and the bicarbonates were converted to methanol. The formate peaks were also decreased first and then increased to a stable state during reaction. Finally, the gas flow was switched to Ar, and the OH* peaks disappeared and other species peaks became weaker and weaker. However, the formate and carbonates could still be observed after purging for 30 min (Figure S18d), indicating that these surface species were stable. In addition, the CO* peaks at 2178 and 2110 cm⁻¹ could not be observed during the transient *in-situ* DRIFTS experiments, thus these CO* species were regarded mainly as spectators during the reaction. Therefore, the surface species of formate, carbonates, bicarbonate, and methoxy are proposed as the main reaction intermediates in the mechanism of CO₂ hydrogenation to methanol.”

In the introduction is discussed that some authors found that Cu-O_v-Ce active species could improve the CO₂ transformation to methanol. The Graciani *et al.* reference did not show that because it deals with model systems composed of CeO_x/Cu(111). Moreover, some authors have proposed that vacancies in Cu/CeO₂ are not beneficial because they are poisoned by adsorbed CO₂ (ACS Catal. 2020, 10, 11532–11544).

Response: We thank the reviewer very much for pointing out this mistake. Graciani *et al.* reference indeed focuses on a copper-ceria interface composed of CeO_x/Cu(111) instead of Cu-O_v-Ce active species¹. **Thus, we have deleted the “Graciani *et al.*” in the introduction section and revised the contents as follows:**

“A photoactivated Cu-CeO₂ catalyst for the preferential oxidation of CO was fabricated through MOFs crystal engineering, which has abundant Cu-O_v-Ce active sites derived from the substitution of Cu into Ce-MOF precursor. Meanwhile, ~~Graciani *et al.*~~ and Yang *et al.* have demonstrated that the copper-ceria solid solution with enhanced Cu-O_v-Ce_x active species could improve the CO₂ hydrogenation to methanol (Page 5)².”

In addition, we highly appreciate some works have proposed a different point that vacancies in Cu/CeO₂ are not beneficial for CO hydrogenation because they are poisoned by adsorbed CO₂ to form carbonate-like species, and thus some authors have developed the chemisorption with improved N₂O titration measurements to determine the amount of oxygen vacancies^{3,4}. Methanol synthesis proceeds predominantly *via* CO₂ hydrogenation on the metallic copper surface. Meanwhile, the mechanistic study indicates that the CO pathway is inhibited by carbonate-like species formed by CO₂ adsorption at the Cu–CeO₂ interface, and the hydrogenation of CO to methanol is inhibited in the presence of CO₂, so CO₂ is the dominant carbon source in CO/CO₂ mixtures. **We have mentioned this point in the discussion of the revised manuscript as depicted below:**

“In addition, it is worth noting that oxygen vacancies in Cu/CeO₂ are not beneficial for CO hydrogenation because they are poisoned by adsorbed CO₂ to form of carbonate-like species. Thus, the roles of oxygen vacancies in CO/CO₂ hydrogenation still need to be illustrated with more researchers’ efforts in the future (Page 33).”

In our study, the calculated results showed that CO₂ prefers to interact with surface oxygen atom to form carbonate on the Cu⁺-CeO_{2-x} and Zn/Cu⁺-CeO_{2-x} catalysts, while those are weakly adsorbed at the Cu-O_v-Ce site with the form of linear CO₂ species. Thus, the oxygen vacancies in the Cu⁺-CeO_{2-x} and Zn/Cu⁺-CeO_{2-x} catalysts would not be poisoned by adsorbed CO₂. From the previous

reports, the ceria crystal face effect on the catalytic performance originates from the difference of the surface atomic arrangement and the electronic properties of CeO₂, which influences the oxygen vacancy formation energy⁵. In a series of reactions, such as CO₂ methanation, CO oxidation and RWGS, the catalytic activity is highly correlated with the number of oxygen vacancies, especially for CO₂ methanation and CO₂ hydrogenation to methanol, in which the number of oxygen vacancies is correlated with activity (**Figure R2**)⁶. Moreover, the number of oxygen vacancies could influence CO₂ activation via lower reaction barriers of CO₂ dissociation on a divacancy as compared with a single isolated oxygen vacancy⁷. We highly appreciated the reviewer mentioning controversial points on the roles of oxygen vacancies in CO₂ hydrogenation, which we have discussed in our manuscript and recommended should be a focus of future research in the field.

Figure R2. The influence of CeO₂ crystal facet and oxygen vacancies on CO₂ hydrogenation to methanol over Pd/CeO₂ catalysts⁶.

References

1. Graciani, J. *et al.* Highly active copper-ceria and copper-ceria-titania catalysts for methanol synthesis from CO₂. *Science* **345**, 546-550 (2014).
2. Yang, B., Deng, W., Guo, L. & Ishihara, T. Copper-ceria solid solution with improved catalytic activity for hydrogenation of CO₂ to CH₃OH. *Chin. J. Catal.* **41**, 1348-1359 (2020).
3. Zhu, J. D. *et al.* Mechanism and nature of active sites for Methanol Synthesis from CO/CO₂ on Cu/CeO₂. *ACS Catal.* **10**, 11532-11544 (2020).
4. Zhu, J. *et al.* Flame synthesis of Cu/ZnO-CeO₂ catalysts: Synergistic metal-support interactions promote CH₃OH selectivity in CO₂ hydrogenation. *ACS Catal.* **11**, 4880-4892 (2021).
5. Zhang, W., Ma, X. L., Xiao, H., Lei, M., & Li, J. Mechanistic investigations on thermal hydrogenation of CO₂ to

- methanol by nanostructured CeO₂(100): The crystal-plane effect on catalytic reactivity. *J. Phys. Chem. C* **123**, 11763–11771 (2019).
6. Jiang, F. *et al.* Insights into the influence of CeO₂ crystal facet on CO₂ hydrogenation to methanol over Pd/CeO₂ catalysts. *ACS Catal.* **10**, 11493–11509 (2020).
7. Kumari, N., Haider, M. A., Agarwal, M., Sinha, N., & Basu, S. Role of reduced CeO₂(110) surface for CO₂ reduction to CO and methanol. *J. Phys. Chem. C* **120**, 16626–16635 (2016).

REVIEWERS' COMMENTS

Reviewer #1 (Remarks to the Author):

Accept.

Reviewer #2 (Remarks to the Author):

This work can be accepted now.

Reviewer #3 (Remarks to the Author):

The responses given by the authors are convincing. I think the new experiments they performed and the changes to the manuscript greatly improved the quality of the work. I recommend publication in this journal.

Responses to Reviewers' Comments

REVIEWERS' COMMENTS

Reviewer #1 (Remarks to the Author):

Accept.

Reviewer #2 (Remarks to the Author):

This work can be accepted now.

Reviewer #3 (Remarks to the Author):

The responses given by the authors are convincing. I think the new experiments they performed and the changes to the manuscript greatly improved the quality of the work. I recommend publication in this journal.

Response: We would like to thank the reviewers for their valuable comments and positive recommendations.